

# The algebra of anomaly interplay

**Joe Davighi$^\star$ and Nakarin Lohitsiri$^\dagger$**

Department of Applied Mathematics and Theoretical Physics,
University of Cambridge, Wilberforce Road, Cambridge, UK

$\star$ jed60@cam.ac.uk,  $\dagger$ nl313@cam.ac.uk

## Abstract

We give a general description of the interplay that can occur between local and global anomalies, in terms of (co)bordism. Mathematically, such an interplay is encoded in the non-canonical splitting of short exact sequences known to classify invertible field theories. We study various examples of the phenomenon in 2, 4, and 6 dimensions. We also describe how this understanding of anomaly interplay provides a rigorous bordism-based version of an old method for calculating global anomalies (starting from local anomalies in a related theory) due to Elitzur and Nair.

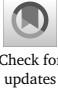
# 1  Introduction

Anomaly inflow relates fermionic anomalies to quantum field theories in one dimension higher. In the perturbative case the anomaly theory has a lagrangian description via Chern–Simons terms, while the non-perturbative generalization involves the exponentiated $\eta$-invariant [1–3] of Atiyah, Patodi, and Singer [4–6]. This idea forms the root of a more general understanding that any anomalous theory can be described by a relative field theory [7] between an extended field theory in one dimension higher, called the anomaly theory, and the trivial extended field theory [8–11]. The anomaly theory is typically a rather special type of quantum field theory, namely an invertible one.[1]

    In many cases the anomaly theory will also be topological. Then the anomaly theory corresponds to a map of spectra. It was proven in [11] that deformation classes of such invertible, topological theories are classified by the torsion subgroup of homotopy classes of such maps.[2] Unfortunately, perturbative anomalies due to massless chiral fermions are not of this type because the Chern–Simons anomaly theory is not strictly topological, having a mild dependence on the background metric [15]. Nonetheless, it is conjectured [11] that a broader class of physically sensible invertible theories (*i.e.* the reflection positive ones), not necessarily topological, are still classified up to deformation by the homotopy classes of maps between spectra – only now non-torsion elements should be included. Such non-torsion elements are required to describe perturbative fermionic anomalies, whose coefficients are in general arbitrary integers. In the case of fermionic anomalies, the torsion elements capture global anomalies (which, in the absence of local anomalies, are described by genuinely topological anomaly theories).

    This formal classification of invertible field theories, and thus of anomaly theories, naturally encodes a possible interplay between global and local anomalies, which is the subject of this paper. While it has become well known that global anomalies in $d$ spacetime dimensions are detected by the torsion subgroup of a bordism group in degree $d + 1$, it is perhaps less widely appreciated that the local anomalies are also detected by bordism invariants, albeit in one degree higher still. The two parts are naturally combined into a dual 'cobordism group'[3] via a short exact sequence, which coincides precisely with the group of homotopy classes of maps of spectra appearing in the classifications of Freed and Hopkins [11].

---

[1]In fact, invertibility is not a strict requirement for an anomaly theory, at least for a certain mathematical definition of anomalies in terms of relative quantum field theory. An example of a non-invertible anomaly theory is provided by the six-dimensional $\mathcal{N} = (2, 0)$ superconformal field theory [9, 12]. However, in such examples the partition function of the original theory is ill-defined even when the background fields that couple to the anomalous symmetry are turned off, and so they do not chime with the usual physics definition of an anomaly. We are content to exclude such examples, and assume invertibility of the anomaly theory in this paper.

[2]Classifying such topological and invertible field theories therefore reduces to computations in stable homotopy theory (see *e.g.* [13, 14] for examples).

[3]We will clarify our usage of the word *co*bordism, as opposed to bordism, in Section 2.

Like the universal coefficient theorem in ordinary cohomology, these short exact sequences defining the cobordism groups split, meaning the most general anomaly 'factors' into its global and local parts, although crucially this splitting is not canonical. This last property allows for an interplay between the global and local anomalies of theories with different symmetries (for example between two gauge theories whose gauge groups are related by some obvious map, such as inclusion of a subgroup). Specifically, local anomalies in one theory can pullback to global anomalies in the other. (The converse is not possible.) Physically, such a pair of theories might be related to each other along an RG flow. This includes, but is not restricted to, the familiar situation whereby only a subgroup of the microscopic symmetries are linearly realised at low energies due to spontaneous symmetry breaking.

Two examples of this interplay were recently observed in $U(2)$ *vs.* $SU(2) \times U(1)$ gauge theories in four dimensions [16], defined with or without spin structures. In the spin case, a bordism computation reveals that the $U(2)$ theory cannot suffer from global anomalies,[4] but nonetheless a $U(2)$ theory with a perturbative anomaly can pullback to an $SU(2)$ theory with a global anomaly. We emphasize that the idea of such an interplay is, however, far from new. For an early example of its use, Elitzur and Nair [17], following Witten [18], showed how the global 4d $SU(2)$ anomaly [19] can be derived from a perturbative anomaly in $SU(3)$ and extended the method to anomalies in higher dimensions. For another well-known example, Ibañez and Ross derived anomalies in discrete $\mathbb{Z}/k$ gauge symmetries, which are necessarily global anomalies, from local anomalies in $U(1)$ [20,21]. It is the recent progress in classifying invertible field theories that allows one to make a precise algebraic statement of such interplay in terms of cobordism, and to see that it is a generic property of the space of anomaly field theories. One application of this formalism is therefore to provide a proper bordism-based version of Elitzur and Nair's suggestion that global anomalies can be precisely derived from perturbative anomalies in some larger group.

In §2 we explain how anomaly interplay is encoded in the non-canonical splitting of a short exact sequence that classifies invertible field theories. As a corollary we describe a method for computing global anomalies (or anomalies in subgroups) using this idea. In the rest of the paper we discuss a number of examples exhibiting the phenomenon of anomaly interplay, ascending the ladder of increasing dimension. We begin by analyzing the interplay between anomalies in $U(1)$ and $\mathbb{Z}/2$ gauge theories in two dimensions, followed by the examples of $U(2)$ gauge theories in four dimensions that were recently analyzed in [16]. We close just as we begin, with an example of anomaly interplay between $U(1)$ and $\mathbb{Z}/2$ gauge theories, but this time in six dimensions. Other examples from recent [22, 23] and not-so-recent [17, 20] references are also discussed from the point of view of our formalism.

## 2 The generalities of anomaly interplay

We are concerned in this paper with anomalies that arise from integrating over massless chiral fermions in $d$ spacetime dimensions, assuming Euclidean signature. The anomaly theory $\mathcal{A}$ is in this case a reflection positive invertible field theory in $d + 1$ dimensions, but not necessarily a topological one. Specifically, its partition function is the exponentiated $\eta$-invariant associated to an appropriate $(d + 1)$-dimensional extension of the Dirac operator, where the $\eta$-invariant is a regularised sum of the signs of the eigenvalues $\lambda_k$ of the Dirac operator. A possible regularization is

$$\eta_X = \lim_{\epsilon \to 0^+} \sum_k e^{-\epsilon |\lambda_k|} \text{sign}(\lambda_k)/2 \,. \tag{2.1}$$

---

[4]Note that $\pi_4(U(2)) \cong \mathbb{Z}/2$ implies there are homotopically non-trivial large gauge transformations in $U(2)$; but the bordism computation means this cannot lead to a global anomaly.

When evaluated on an open $(d+1)$-manifold $X$, the phase $\exp 2\pi i \eta_X$ equals the phase of the anomalous partition function living on the $d$-dimensional boundary $\partial X$ [3]. If $\exp 2\pi i \eta$ equals unity on all closed $d+1$ manifolds to which the necessary structures are extended, then there is no anomaly. In the case where there are only global anomalies, the anomaly theory is strictly topological; in fact $\exp 2\pi i \eta$ becomes bordism invariant under these conditions, which is stronger than topological invariance.

Such invertible field theories in $n := d+1$ dimensions fit inside a formal classification in terms of maps of spectra, conjectured by Freed and Hopkins in Ref. [11]. Before we discuss the physics of anomaly interplay, we must first recap some technicalities of their conjecture. The classification of invertible field theories depends on two pieces of data; the spacetime dimension $n$ and the symmetries of the theory. Following [11], the symmetry type $(H_n, \rho_n)$ of an Euclidean quantum field theory consists of a compact Lie group $H_n$ and a homomorphism

$$\rho_n : H_n \to O(n), \tag{2.2}$$

whose image (either $O(n)$ or $SO(n) \subset O(n)$) constitutes the Wick-rotated spacetime symmetries, and whose kernel $K$ defines the internal symmetries of the theory.[5] Given a fixed symmetry type $(H_n, \rho_n)$, the deformation classes $[\cdot]_{\text{def}}$ of reflection positive, invertible extended field theories in $n$ spacetime dimensions are classified by homotopy classes of maps of spectra [11, Conjecture 8.37],

$$[\mathcal{A}]_{\text{def}} \in [MTH, \Sigma^{n+1} I\mathbb{Z}]. \tag{2.3}$$

Here, the source spectrum $MTH$ is the Madsen–Tillman spectrum associated to a stabilization $H$ of the symmetry type $H_n$. This stabilization is a dimension-independent way of describing the symmetry type of the theory, which is technically defined to be the colimit of a sequence of compact Lie groups $H_m$ (for all $m > n$) that fit inside a commutative diagram of group homomorphisms,

$$
\begin{array}{ccccccc}
H_n & \xrightarrow{i_n} & H_{n+1} & \xrightarrow{i_{n+1}} & H_{n+2} & \xrightarrow{i_{n+2}} & \cdots \\
\rho_n \downarrow & & \rho_{n+1} \downarrow & & \rho_{n+2} \downarrow & & \\
O(n) & \longrightarrow & O(n+1) & \longrightarrow & O(n+2) & \longrightarrow & \cdots ,
\end{array}
\tag{2.4}
$$

where all the horizontal arrows denote injections, and the squares are pullbacks. This sequence of symmetry groups can be used to construct a sequence of Madsen–Tillman spectra $MTH_n$ whose colimit is the spectrum $MTH$ that appears in (2.3). For the mathematical definitions of all these objects, we are content to refer the reader to [11]. For our purposes, it is important to emphasize that the object $MTH$ encodes the specific information about the symmetries of the theory.

The target spectrum $\Sigma^{n+1} I\mathbb{Z}$ that appears in (2.3), on the other hand, is a universal object, in particular a suspension shift of the Anderson dual $I\mathbb{Z}$ of the sphere spectrum. Reflection positivity is required for the corresponding Lorentzian field theory to be unitary.

The massless chiral fermion anomaly theories that we are interested in fit inside this classification, with $n = d+1$ and symmetry $H$ – although we should stress that not every such deformation class in $[MTH, \Sigma^{d+2} I\mathbb{Z}]$ can be realised as the anomaly theory from free chiral fermions.[6] In other words, if we cut a $(d+1)$-dimensional bulk theory with such an invertible phase, then we are not guaranteed a boundary theory of free chiral fermions in $d$ dimensions.

---

[5]The assumption of compactness means that this formalism cannot account for non-compact internal symmetries or supersymmetries. Nor does the symmetry type take into account higher-form symmetries.

[6]In particular [11, Conjecture 9.70], the free fermion anomalies correspond to a subset of maps of spectra from $MTH$ to $\Sigma^{n+1} I\mathbb{Z}$, namely those that factor through a sequence of $KO$-theory groups.

## Exact sequences of anomaly theories

The universal target spectrum $I\mathbb{Z}$ represents a particular generalized cohomology theory, call it $H^\bullet_{I\mathbb{Z}}$. In cohomological language, a deformation class of anomaly theory, for symmetry type $H$ and original spacetime dimension $d$ (of the anomalous fermionic theory), is then a degree $d+2$ class in the $H^\bullet_{I\mathbb{Z}}$ cohomology of the spectrum $MTH$. As described in [10, 11], this generalized cohomology group sits inside a short exact sequence, analogous to the universal coefficient theorem for ordinary cohomology,

$$0 \to \mathrm{Ext}^1(\pi_{d+1}(MTH), \mathbb{Z}) \longrightarrow H^{d+2}_{I\mathbb{Z}}(MTH) \longrightarrow \mathrm{Hom}(\pi_{d+2}(MTH), \mathbb{Z}) \to 0, \qquad (2.5)$$

where $\pi_k$ denotes stable homotopy groups. This exact sequence will be central to our discussion, since it determines how a general anomaly is built out of both global and local parts. The exact sequence splits, though not canonically (as we discuss soon).

Let us unpack this short exact sequence in a special case of particular interest, namely where fermions are defined using a spin structure and charged under gauge group $G$.[7] In this case, a suitable stabilization of the symmetry type is

$$H = \mathrm{Spin} \times G, \qquad (2.6)$$

where $\mathrm{Spin} = \mathrm{colim}_{n\to\infty} \mathrm{Spin}(n)$, and the stable homotopy groups that appear in (2.5) coincide with bordism groups,

$$\pi_k(MTH) \cong \Omega^{\mathrm{Spin}}_k(BG). \qquad (2.7)$$

For this reason it makes sense to refer to the generalized *co*homology theory $H^\bullet_{I\mathbb{Z}}$ as a *co*bordism theory. In this paper we therefore reserve the word 'cobordism' to refer only to the (Anderson) dual of bordism, a terminology which is not standard in the mathematical literature (wherein the generalized homology theory that we exclusively refer to as 'bordism' was originally introduced under the name 'cobordism').

Now, the left factor of the exact sequence (2.5) can here be written as

$$\mathrm{Hom}(\mathrm{Tor}\,\Omega^{\mathrm{Spin}}_{d+1}(BG), \mathbb{R}/\mathbb{Z}). \qquad (2.8)$$

This group detects global anomalies because, as previously mentioned, the exponentiated $\eta$-invariant that appears as the phase of the fermionic partition function [3] becomes a bordism invariant when perturbative anomalies vanish [1–3, 24, 25], and the global anomalies will be of finite order.

To understand how local anomalies are captured by the cobordism group in (2.5), consider now the case where $\Omega^{\mathrm{Spin}}_{d+1}(BG)$ vanishes (which means there is no possible global anomaly). The anomaly theory can then be computed on any closed $(d+1)$-manifold $X$ using the Atiyah–Patodi–Singer (APS) index theorem [4–6] on a $(d+2)$-manifold $Y$ whose boundary is $X$, and to which the $\mathrm{Spin} \times G$ structure extends. We have $\exp 2\pi i \eta(X) = \exp 2\pi i \int_Y \Phi_{d+2}$, where $\Phi_{d+2}$ is the anomaly polynomial (which coincides, locally, with the exterior derivative of the Chern–Simons form in degree $d+1$). This corresponds to the image of the deformation class of this anomaly theory in the group

$$\mathrm{Hom}(\Omega^{\mathrm{Spin}}_{d+2}(BG), \mathbb{Z}), \qquad (2.9)$$

which appears as the right factor of the exact sequence (2.5) defining the cobordism group $H^{d+2}_{I\mathbb{Z}}(MT(\mathrm{Spin} \times G))$. (Note that $\int_Y \Phi_{d+2}$ is constant on bordism classes simply by virtue of Stokes' theorem, and so does provide a well-defined map out of $\Omega^{\mathrm{Spin}}_{d+2}(BG)$.) Exactness of the

---

[7]The examples in §§4, 5.1, 6, 7.1, and 8 fall within this class. The examples in §§5.2 and 7.2 do not, however the generalization of the corresponding bordism groups is straightforward.

sequence in the middle means that, when the local anomalies vanish, there may still be non-trivial global anomalies and that these correspond precisely to the image of the (injective) map from the left factor into $H_{I\mathbb{Z}}^{d+2}(MTH)$.

Before continuing, it is important to remark that the generalized cohomology groups $H_{I\mathbb{Z}}^{d+2}(MTH)$ classify only the deformation classes of anomaly theories, not (isomorphism classes of) anomaly theories themselves. Said more prosaically, the right factor of the exact sequence (2.5) can only measure rather coarse information about the local anomaly, namely the anomaly coefficients, and not the differential form $\Phi_{d+2}$ itself. Recall that the fermionic anomaly theory $\exp 2\pi i \eta$ is a section of the inverse determinant line bundle associated to the Dirac operator [2]. The perturbative anomaly can be calculated by taking a holonomy of the determinant line bundle around a contractible loop in the parameter space (see *e.g.* [26] for one account of this perspective). The fully non-perturbative factor of $\exp 2\pi i \eta$ arises more generally from the holonomy around *any* given loop over the parameter space, and so can capture both global and local anomalies. The complete information describing the anomaly theory is thus encoded in the holonomies of a principal line bundle over the parameter space, equivalently in a principal line bundle *with connection* (up to isomorphism). Therefore one really needs to use a *differential* cohomology theory, specifically a differential refinement of $H_{I\mathbb{Z}}^{\bullet}(MTH)$,[8] to describe the local anomalies properly. Nonetheless, for the purpose of analysing the interplay between global and local anomalies, we find that the 'topological' theory $H_{I\mathbb{Z}}^{\bullet}(MTH)$ will suffice.

## Anomaly interplay as non-canonical splitting

We have seen that the short exact sequence (2.5) tells us how a general fermionic anomaly is put together out of global and local pieces, each of which are classified (up to deformation class) using bordism data. We now discuss the interplay between local and global anomalies in related theories.

To do so, we must first explain what exactly we mean by 'related' theories. We have seen that deformation classes of anomalies are essentially determined by two inputs, a spectrum $MTH$ (given a stabilization $H$ of the symmetry type) and a spacetime dimension. It is therefore natural to introduce "morphisms" between theories of the same dimension by specifying appropriate maps between two underlying symmetry types $H$ and $H'$. Now, maps of spectra constitute the usual notion of morphisms in the category of spectra, which is the appropriate domain for the cohomology functor $H_{I\mathbb{Z}}^{\bullet}$. So an appropriate morphism between theories ought to correspond to a map of spectra $\pi : MTH \to MTH'$. One way to construct such a map of spectra is to provide a sequence of group homomorphisms $\pi_n : H_n \to H'_n$ on the underlying symmetry types, which commute with the homomorphisms $(i_n, \rho_n)$ of Eq. (2.4). This induces maps between the spaces of the Madsen–Tillman spectra that commute with the structure maps, thus giving a function between spectra *ergo* a map of spectra.

In special cases such as (2.6) where the symmetry type factors into a product of a spacetime symmetry and an internal symmetry that is independent of the dimension $n$, and where $H$ and $H'$ share the same spacetime symmetry, one can replace the set of homomorphisms $\pi_n$ by a single homomorphism $\pi$ between the internal symmetry groups. This suggests a convenient abuse of notation, in which we frequently use the shorthand $\pi : H \to H'$ to denote the set of homomorphisms on the underlying $H$-structures that give rise to the map of spectra $\pi : MTH \to MTH'$.

Continuing, given a pair of symmetry types $H$ and $H'$ and such a $\pi : H \to H'$, there is a pullback $\pi^* : H_{I\mathbb{Z}}^{\bullet}(MTH') \to H_{I\mathbb{Z}}^{\bullet}(MTH)$ between anomaly theories. Indeed, there is a

---

[8]This differential refinement could be constructed using the tools set out in [27].

pullback diagram for the whole short exact sequence (2.5),

$$0 \longrightarrow \mathrm{Ext}^1(\pi_{d+1}(MTH),\mathbb{Z}) \longrightarrow H_{I\mathbb{Z}}^{d+2}(MTH) \longrightarrow \mathrm{Hom}(\pi_{d+2}(MTH),\mathbb{Z}) \longrightarrow 0$$

$$\pi^* \uparrow \qquad\qquad \pi^* \uparrow \qquad\qquad \pi^* \uparrow \qquad\qquad\qquad (2.10)$$

$$0 \longrightarrow \mathrm{Ext}^1(\pi_{d+1}(MTH'),\mathbb{Z}) \longrightarrow H_{I\mathbb{Z}}^{d+2}(MTH') \longrightarrow \mathrm{Hom}(\pi_{d+2}(MTH'),\mathbb{Z}) \longrightarrow 0\,,$$

which is a commutative diagram. Like the universal coefficient theorem in ordinary cohomology, the short exact sequence for $H_{I\mathbb{Z}}^\bullet$ splits, but the splitting is not canonical. This means that, while there exist splitting maps for each row of (2.10), *i.e.* homomorphisms going horizontally from right to left, these splitting maps do not give rise to commutative squares. We remark that the same pullback diagram (2.10) was recently used in [28] to study gauged Wess–Zumino–Witten terms from a bordism perspective.

In many of the ensuing examples, we will be interested in the special case of (2.10) where both symmetry types $H$ and $H'$ involve a spin structure, as in (2.6). Then the anomaly interplay diagram can be written in terms of spin bordism groups as

$$0 \longrightarrow \mathrm{Hom}(\mathrm{Tor}\,\Omega_{d+1}^{\mathrm{Spin}}(BG),\mathbb{R}/\mathbb{Z}) \longrightarrow H_{I\mathbb{Z}}^{d+2}(MTH) \longrightarrow \mathrm{Hom}(\Omega_{d+2}^{\mathrm{Spin}}(BG),\mathbb{Z}) \longrightarrow 0$$

$$\pi^* \uparrow \qquad\qquad \pi^* \uparrow \qquad\qquad \pi^* \uparrow \qquad\qquad\qquad (2.11)$$

$$0 \longrightarrow \mathrm{Hom}(\mathrm{Tor}\,\Omega_{d+1}^{\mathrm{Spin}}(BG'),\mathbb{R}/\mathbb{Z}) \longrightarrow H_{I\mathbb{Z}}^{d+2}(MTH') \longrightarrow \mathrm{Hom}(\Omega_{d+2}^{\mathrm{Spin}}(BG'),\mathbb{Z}) \longrightarrow 0,$$

where the various bordism groups can often be straightforwardly computed using, say, the Atiyah–Hirzebruch or the Adams spectral sequence.

To illustrate how this non-canonical splitting might manifest itself, we will frequently encounter scenarios where the pullback of exact sequences takes the following form

$$0 \longrightarrow \mathrm{Global} \longrightarrow H_{I\mathbb{Z}}^{d+2}(MTH) \longrightarrow 0 \longrightarrow 0$$

$$\uparrow \qquad\qquad \pi^* \uparrow \qquad\qquad \uparrow \qquad\qquad\qquad (2.12)$$

$$0 \longrightarrow 0 \longrightarrow H_{I\mathbb{Z}}^{d+2}(MTH') \longrightarrow \mathrm{Local}' \longrightarrow 0$$

in which a non-zero element in Local$'$, corresponding to a local anomaly in the second theory, can pullback to a non-zero element in $H_{I\mathbb{Z}}^{d+2}(MTH)$ (by first following the splitting map to the left, then pulling back the anomaly theory along $\pi^*$ in the middle column), which must correspond to a global anomaly in the first theory.[9] We will study many examples of this scenario in this paper, for example in §5 in which $H = \mathrm{Spin} \times SU(2)$ (which has only global anomalies for $d = 4$) and $H' = \mathrm{Spin} \times U(2)$ (which has only local anomalies), with $\pi : H \to H'$ defining the usual embedding of $SU(2)$ as a subgroup of $U(2)$.

Importantly, the 'reverse' situation in which a global anomaly pulls back to a local anomaly is not possible. Suppose the theory $H'$ theory has only global anomalies, and the $H$ theory has only local anomalies, where again $\pi : H \to H'$ denotes a group homomorphism. There is a commutative diagram

$$0 \longrightarrow 0 \longrightarrow H_{I\mathbb{Z}}^{d+2}(MTH) \longrightarrow \mathrm{Local} \longrightarrow 0$$

$$\uparrow \qquad\qquad \pi^* \uparrow \qquad\qquad \uparrow \qquad\qquad\qquad (2.13)$$

$$0 \longrightarrow \mathrm{Global}' \longrightarrow H_{I\mathbb{Z}}^{d+2}(MTH') \longrightarrow 0 \longrightarrow 0.$$

---

[9]We emphasize that, were the splitting canonical, the right-hand square of (2.12) with its horizontal arrows reversed would be commutative, which would imply the composite map just described were the zero map. Thus if the split were canonical there would be no possibility of anomaly interplay.

Here the anomaly pullback map $\pi^*$ must be the zero map, forbidding any anomaly interplay, because $\pi^*$ is a homomorphism from a finite abelian group into a free one, and thus zero. For a physics example, consider embedding $H = \text{Spin} \times U(1)$, which has only local anomalies, as the Cartan of $H' = \text{Spin} \times SU(2)$, which has only global anomalies. Any $SU(2)$ theory with a global anomaly necessarily pulls back to a $U(1)$ theory that is free of local anomalies; an $SU(2)$ doublet, say, decomposes to a pair of opposite charged particles under the Cartan $U(1)$ subgroup.

Thus, completely generally, the possibilities for pulling back a local or global anomaly are

$$\text{Local anomalies} \xrightarrow{\text{pullback}} \text{Local and/or global anomalies,}$$
$$\text{Global anomalies} \xrightarrow{\text{pullback}} \text{Global anomalies only.}$$

The first option corresponds to what we call 'anomaly interplay'.

A crucial step in this analysis of anomalies is the pulling back of a particular anomaly theory, which corresponds to finding the pullback map $\pi^* : H^\bullet_{I\mathbb{Z}}(MTH') \to H^\bullet_{I\mathbb{Z}}(MTH)$. Precisely because of anomaly interplay, this cannot simply be achieved by pulling back the local and global anomalies separately and then combining the two via a Cartesian product. Alas, we will not in this paper give a general 'formula' for this pullback map for any pair of symmetry types. However, in specific cases we often find that it can be computed using simple arguments.

Finding the anomaly pullback map turns out to be especially simple in a rather important special case, namely where

1. $H$ is a subgroup of $H'$, with $\pi : H \to H'$ the embedding,

2. the $H'$ theory has only local anomalies and the $H$ theory has only global anomalies, as indicated by Eq. (2.12), and

3. the dimension $d$ of the original anomalous theory is even.

$$(2.14)$$

Most of the examples we examine in this paper fall into this class (see §§ 5, 6, and 7).

Computing the pullback map between anomaly theories is simple under these conditions in large part because it becomes straightforward (in either theory) to explicitly evaluate the anomaly theory on an arbitrary closed $(d + 1)$-manifold (with appropriate $H$-structure). For the theory with only local anomalies, the vanishing of $\text{Hom}(\text{Tor}\,\Omega^{H'}_{d+1}, \mathbb{R}/\mathbb{Z})$ implies $\Omega^{H'}_{d+1} = 0$ because, in odd degrees, such spin bordism groups are pure torsion. Therefore, the APS index theorem can be used to evaluate $\exp 2\pi i \eta_X = \exp 2\pi i \int_Y \Phi_{d+2}$ for any closed $(d + 1)$-manifold $X = \partial Y$, for any potentially-anomalous fermion representation $\mathbf{R}'$ of $H'$. To pullback this anomaly theory,[10] it suffices to decompose into representations of the subgroup, *viz.* $\mathbf{R}' \to \bigoplus_\alpha \mathbf{R}^\alpha$, and then evaluate $\exp 2\pi i \eta_X$ on all closed $(d + 1)$-manifolds with $H$ structure for the representations $\mathbf{R}^\alpha$. This is in principle straightforward given the assumption that $H$ is free of local anomalies, because in that case $\exp 2\pi i \eta_X$ is constant on bordism classes, and so we need only evaluate $\exp 2\pi i \eta_X$ on each generator of the (finitely-generated) bordism group $\Omega^H_{d+1}$, for each $\mathbf{R}^\alpha$. This last step is often easier said than done; however, in the particular examples in §§ 5, 6, and 7, we will for the most part get away with using known results.

---

[10]The observant reader will quite rightly object that we have not strictly pulled back the anomaly theory just by evaluating it on closed top manifolds – indeed we also need to evaluate the theory on manifolds of codimension 1, codimension 2, and so on, to fully specify the extended field theory.

# 3 Physics applications

## 3.1 Deriving global anomalies

The procedure just described for pulling back the anomaly theory can be turned around to give a new method for computing global anomalies, and the conditions for their cancellation, starting from purely local anomalies in a '$\pi$-related' theory. We envisage this as being an important application of our rather formal treatment of anomaly interplay.

The general idea is as follows. Suppose we identify a symmetry type $H$ for which the bordism group $\Omega^H_{d+1}$ is non-vanishing torsion, giving the possibility of a global anomaly. To compute the global anomaly one must evaluate the $\eta$-invariant on each generator of $\Omega^H_{d+1}$, which is likely a hard task. But suppose one can embed $H$ as a subgroup of some $H'$, and for simplicity let us assume that the set of conditions (2.14) hold. This induces an injection $\pi_* : \mathcal{M}^H_{d+1} \to \mathcal{M}^{H'}_{d+1}$ from a set of closed $(d+1)$-manifolds with $H$ structure $\mathcal{M}^H_{d+1}$ to a set of closed $(d+1)$-manifolds with $H'$ structure $\mathcal{M}^{H'}_{d+1}$. Let $X \in \mathcal{M}^H_{d+1}$ be a representative of a generator of $\Omega^H_{d+1}$. By assumption (2.14), $\pi_* X$, whose $H$ structure is viewed as an $H'$ structure, is nullbordant, *i.e.* the boundary of a $(d+2)$-manifold $Y$ to which the $H'$ structure extends. One can then evaluate on $\pi_* X$ the anomaly theory for a representation $\mathbf{R}'$, which we pick to correspond to a generator of the group $\mathrm{Hom}(\Omega^{H'}_{d+2}, \mathbb{Z})$ classifying local anomalies, using the APS index theorem, *viz.* $\exp 2\pi i \eta(\pi_* X, \mathbf{R}') = \exp 2\pi i \int_Y \Phi_{d+2}[\mathbf{R}']$. Since we can view $\eta$ as a function (on its first argument) from $\mathcal{M}^{H'}_{d+1}$ to $\mathbb{R}$, its pullback $\pi^* \eta$, which is the eta-invariant for the $H$ theory, is defined as $\pi^* \eta = \eta \circ \pi_*$. Therefore, $\exp 2\pi i \int_Y \Phi_{d+2}[\mathbf{R}']$ must equal $\exp(2\pi i(\pi^* \eta)(X, \bigoplus_\alpha \mathbf{R}^\alpha)) = \prod_\alpha \exp(2\pi i(\pi^* \eta)(X, \mathbf{R}^\alpha))$. Thus, if the phase $\exp(2\pi i \int_Y \Phi_{d+2}[\mathbf{R}']) = \exp 2\pi i/k \neq 1$, then there is a global anomaly in the original $H$ theory for the set of representations $\mathbf{R}^\alpha$. Moreover, if the phase $\exp 2\pi i/k$ is of maximal order in $\Omega^H_{d+1}$ then we have identified a generator of the global anomaly, and can derive the general conditions for cancelling the global anomaly.

Although described abstractly here, we will put to work this method for deriving global anomalies in §§4–8. For example, we will see how the general condition for cancelling the 4d $SU(2)$ global anomaly can be derived by computing only local anomalies in either $U(2)$ (§5) or $SU(3)$ (§6). In the 2d example of §4 we will use this method to evaluate the exponentiated $\eta$-invariant on a generator of the bordism group $\Omega^{\mathrm{Spin}}_3(B\mathbb{Z}/2) \cong \mathbb{Z}/8$ from scratch, by embedding $\mathbb{Z}/2$ inside $U(1)$ and thence using only perturbative anomalies.

The idea to derive global anomalies from perturbative anomalies in larger groups was first proposed as a method for analyzing global anomalies by Elitzur and Nair in [17],[11] following Witten [18]. However, that paper preceded the modern popularization of the use of bordism invariants to study anomalies,[12] and certainly preceded widespread knowledge of the exact sequence (2.5) used to classify reflection positive invertible field theories which is crucial to our arguments here. Instead, the main algebraic tool used in [17] was an exact sequence of homotopy groups (with spacetime accordingly assumed to have spherical topology[13]). This was applied, for example, to derive the 4d $SU(2)$ global anomaly for an $SU(2)$ doublet, evaluated on a spacetime $M \cong S^4$, from the perturbative anomaly for the triplet representation of $SU(3)$. In §6 we recast this analysis using the bordism-based version of anomaly interplay set out in this paper. This bordism-based method, which results in a condition on the anomaly

---

[11]The homotopy-based method of [17] has also been applied more recently to analyze global anomalies in $d = 8$ [29].

[12]It is worth emphasizing that the crucial observations regarding bordism are not so modern, going back almost as far as Elitzur and Nair's homotopy-based work, to Ref. [1].

[13]No such requirement is made in the bordism version which, in accordance with locality, allows for arbitrary spacetime topology.

polynomial reduced mod 2 (via the APS index theorem), is powerful enough to derive the full condition for cancelling the $SU(2)$ anomaly on any manifold, and given arbitrary fermion content. (The condition is that the total number of $SU(2)$ multiplets with isospins $j \in 2\mathbb{Z}_{\geq 0} + 1/2$ should be even.) Moreover, the method is arguably more straightforward, with no need to consider homotopy classes of specific gauge field configurations.

From this perspective, one purpose of the present paper is to give a rigorous bordism-based version of Elitzur and Nair's method for computing global anomalies, or more generally for computing any anomalies in subgroups.

We also want to stress that, despite giving the correct results in various examples, computing homotopy groups does not offer a direct way to detect global anomalies, which are correctly detected by bordism (in one degree higher). Thus, it is perhaps not surprising that homotopical methods have been erroneously applied to postulate various 6d global anomalies, for $G = SU(2)$, $SU(3)$, and $G_2$.[14] (A quick bordism calculation reveals that none of these theories can in fact suffer global anomalies.)

## 3.2 Anomaly matching

We have described one use of anomaly interplay as a technical method for computing global anomalies from anomaly polynomials, in which the larger gauge group $H'$ is a mathematical device. From a physics perspective, the maps $\pi^*$ which we use to pullback anomaly theories can often be interpreted in terms of renormalization group (RG) flows. This will be the case in several of the examples we consider in the rest of this paper. When a quantum field theory flows under RG there are various ways in which the symmetry type can change. The most well-known is that some symmetries may become spontaneously broken at low energies, typically leading to massless Goldstone bosons if the broken symmetry were global, and gauge bosons acquiring mass in the case of spontaneously broken gauge symmetries. There are, however, more exotic possibilities; for example, certain theories feature a global symmetry enhancement at high energies (famously this occurs in 5d supersymmetric gauge theories, as originally conjectured in [33]). The changing symmetry type could also involve the spacetime symmetry structures; for example, coupling a Spin-$SU(2)$ theory to certain Higgs fields can trigger an RG flow that dynamically generates a spin structure in the IR [34].

In the case of spontaneous symmetry breaking, the IR will exhibit a subgroup $G_{\text{IR}} \subset G_{\text{UV}}$ of the UV symmetries. There is always an injective homomorphism $\pi : G_{\text{IR}} \to G_{\text{UV}}$ defining the embedding of a subgroup, which can be used to analyze anomalies (and their interplay) along this RG flow.[15] The pull-back maps $\pi^*$ in (2.10) then go in the other direction, and so determine a homomorphism from the full anomaly in $G_{\text{UV}}$ to the anomalies in the subgroup $G_{\text{IR}}$. By the arguments above, this generically involves an interplay between local and global anomalies. In the case that (2.10) describes 't Hooft anomalies in global symmetries, in which the RG flow is between consistent quantum field theories, the constraint of anomaly matching then means that any residual anomalies must be matched by the bosonic degrees of freedom that emerge in the IR, via Wess–Zumino–Witten terms. The non-perturbative version of this mechanism, which is needed in the present context to include any global anomalies, was discussed in [35].

---

[14]The absence of these 6d anomalies is discussed in Refs. [30, 31], building on [32]. More generally, the role of homotopy groups in detecting global anomalies has been re-examined in [30].

[15]In rare cases there might also exist a group homomorphism going the other way, *viz.* $\pi : G_{\text{UV}} \to G_{\text{IR}}$. For example, if $G_{\text{UV}}$ is a finite abelian group, then any subgroup of IR symmetries can be realised as the image of a homomorphism. Given the pullbacks $\pi^*$ go in the opposite direction, this could in principle allow one to determine a finite UV anomaly from the finite anomaly in the IR. If $G$ is a simple group, however, then the image of any homomorphism out of $G$ is either zero or all of $G$.

# 4   2d anomalies in $U(1)$ *vs.* $\mathbb{Z}/2$

For our first concrete example, we discuss the anomaly interplay between $\mathbb{Z}/2$ and $U(1)$ gauge theories in two dimensions, each defined using a spin structure. Let the map $\pi : \mathbb{Z}/2 \to U(1)$ denote the usual embedding of $\mathbb{Z}/2 \subset U(1)$ which maps the non-trivial element of $\mathbb{Z}/2$ to $e^{i\pi}$. This naturally induces the map $\pi : H = \mathrm{Spin} \times \mathbb{Z}/2 \to H' = \mathrm{Spin} \times U(1)$ between symmetry types.

**Bordism account of the anomalies**

To compute the short exact sequences (2.5) that capture all possible anomalies for these theories in two dimensions, we need to compute the torsion subgroup of $\Omega_3$ and the free part of $\Omega_4$ for each symmetry type. The relevant bordism groups for the $H$ theory are[16]

$$\Omega_3^{\mathrm{Spin}}(B\mathbb{Z}/2) \cong \mathbb{Z}/8, \quad \Omega_4^{\mathrm{Spin}}(B\mathbb{Z}/2) \cong \mathbb{Z}, \tag{4.1}$$

leading to anomalies classified by the short exact sequence

$$0 \to \mathbb{Z}/8 \to \mathbb{Z}/8 \times \mathbb{Z} \to \mathbb{Z} \to 0. \tag{4.2}$$

Unlike many of the examples of anomaly interplay that will be discussed later on, the cobordism group here detects both a global and a local anomaly. However, this particular local anomaly is already present in a fermionic system without any internal symmetry. This can be seen from the fact that there is a canonical splitting $\Omega_4^{\mathrm{Spin}}(B\mathbb{Z}/2) \cong \Omega_4^{\mathrm{Spin}}(\mathrm{pt}) \oplus \tilde{\Omega}_4^{\mathrm{Spin}}(B\mathbb{Z}/2)$, where the second direct summand is the reduced bordism group, and $\Omega_4^{\mathrm{Spin}}(\mathrm{pt}) \cong \mathbb{Z}$, a generator for which is the K3 surface. The dual factor of $\mathbb{Z}$ appearing in (4.2) is therefore a pure gravitational anomaly. Indeed, we know that without any extra symmetry the 4$^{\mathrm{th}}$ anomaly polynomial $\Phi_4$ does not vanish, but is given by $-p_1/24$, where $p_1$ is the first Pontryagin class of the tangent bundle. One may identify the anomaly theory with the 3d gravitational Chern–Simons term.

On the other hand, the relevant bordism groups for $H'$ are

$$\Omega_3^{\mathrm{Spin}}(BU(1)) = 0, \quad \Omega_4^{\mathrm{Spin}}(BU(1)) \cong \mathbb{Z} \times \mathbb{Z}. \tag{4.3}$$

A general local anomaly is thus classified by two integers, which in this case one can take to be just the $U(1)$ anomaly coefficient $\mathcal{A}_{\mathrm{gauge}}$ and the pure gravitational anomaly coefficient $\mathcal{A}_{\mathrm{grav}}$. Consider an arbitrary spectrum of charged fermions, consisting of a set of $N_L$ left-moving Weyl fermions with charges $q_1, \ldots, q_{N_L}$ together with $N_R$ right-moving Weyls with charges $r_1, \ldots, r_{N_R}$. The anomaly coefficients are given by

$$\mathcal{A}_{\mathrm{gauge}} = \sum_{i=1}^{N_L} q_i^2 - \sum_{i=1}^{N_R} r_i^2, \tag{4.4}$$
$$\mathcal{A}_{\mathrm{grav}} = N_L - N_R.$$

Note that in 2 dimensions, unlike in 4, conjugating a complex fermion does not flip its chirality; hence we cannot now take all the Weyls to be left-moving without loss of generality.

---

[16]We remark that this $\mathbb{Z}/8$-valued global anomaly, for a 2d theory with unitary $\mathbb{Z}/2$ symmetry, is related to a parity anomaly for (0+1)-dimensional Majorana fermions [36]. This can be understood in terms of the Smith isomorphism $\Omega_3^{\mathrm{Spin}}(B\mathbb{Z}/2) \cong \Omega_2^{\mathrm{Pin}^-} \cong \mathbb{Z}/8$ [37] (see also [38]).

**The anomaly interplay**

We now study the anomaly interplay between $\mathbb{Z}/2$ and $U(1)$ gauge theories in 2d. As usual, the subgroup embedding $\pi : \mathbb{Z}/2 \to U(1)$ induces a map of spectra $\pi : MTH \to MTH'$ as well as a pullback diagram for the anomaly theories,

$$
\begin{array}{ccccccccc}
0 & \longrightarrow & \mathbb{Z}/8 & \longrightarrow & \mathbb{Z}/8 \times \mathbb{Z} & \longrightarrow & \mathbb{Z} & \longrightarrow & 0 \\
 & & \uparrow & & {\pi^*}\uparrow & & \uparrow & & \\
0 & \longrightarrow & 0 & \longrightarrow & \mathbb{Z} \times \mathbb{Z} & \longrightarrow & \mathbb{Z} \times \mathbb{Z} & \longrightarrow & 0,
\end{array} \tag{4.5}
$$

which encodes an anomaly interplay between a $U(1)$ local anomaly and a global anomaly in the $\mathbb{Z}/2$ theory through the pullback $\pi^*$. The gravitational anomaly, corresponding to the second factor in $\Omega_4^{\mathrm{Spin}}(BU(1)) \cong \mathbb{Z} \times \mathbb{Z}$, maps to itself under $\pi^*$, playing no role in the interplay. This will always be the case for pure gravitational anomalies whenever we consider a pair $H$ and $H'$ of symmetry types that differ only in their internal symmetry groups.

To work out how the pullback acts on the first factor, we consider the generic fermion spectrum coupled to a 2d $U(1)$ gauge field described above, and begin by making some simple arithmetic observations. To wit, let the $N_L$ left-moving Weyl fermions split into $N_L^e$ fermions with even charges $2k_i$, $i = 1, \ldots, N_L^e$ and $N_L^o$ fermions with odd charges $2l_i + 1$, $i = 1, \ldots, N_L^o$. Similarly, let the $N_R$ right-moving Weyl fermions divide into $N_R^e$ fermions with even charges $2k_i'$, $i = 1, \ldots, N_R^e$, and $N_R^o$ fermions with odd charges $2l_i' + 1$, $i = 1, \ldots, N_R^o$. The gravitational anomaly cancellation condition requires that the index $N_L - N_R$ vanishes. In terms of our variables, this reads

$$
(N_L^e - N_R^e) + (N_L^o - N_R^o) = 0 \,. \tag{4.6}
$$

The $U(1)$ gauge anomaly cancels when $\mathcal{A}_{\mathrm{gauge}} = 0$, which in these variables translates to the condition

$$
4 \left[ \sum_{i=1}^{N_L^e} k_i^2 - \sum_{i=1}^{N_R^e} k_i'^2 + \sum_{i=1}^{N_L^o} l_i(l_i + 1) - \sum_{i=1}^{N_R^o} l_i'(l_i' + 1) \right] + N_L^o - N_R^o = 0 \,. \tag{4.7}
$$

The second condition immediately implies that the index for the oddly-charged Weyl fermions must be a multiple of 4, *viz.*

$$
N_L^o - N_R^o \in 4\mathbb{Z} \,. \tag{4.8}
$$

Now consider a $\mathbb{Z}/2$ subgroup of this $U(1)$ gauge group. It acts on a fermion with charge $q$ as $(-1)^q$. In other words, fermions with even $U(1)$ charges decouple from this $\mathbb{Z}/2$ gauge field, and only the oddly-charged fermions can contribute to the $\mathbb{Z}/8$ global anomaly encoded in (4.2). Since the anomaly pullback map $\pi^*$ is a homomorphism it maps zero to zero, and thus maps any anomaly-free spectrum to another. So if a single left-moving, $\mathbb{Z}/2$-charged Weyl fermion contributes an anomaly equal to $\nu \bmod 8$, then our considerations in the previous paragraph imply that $4\nu = 0 \bmod 8$. Thus, the elementary algebra above tells us that either $\nu = 0$, 2 or 4 mod 8. To see which of these maps is the correct one, we should evaluate the $\eta$-invariant on a generator of the appropriate bordism group for a single $\mathbb{Z}/2$-charged Weyl fermion. The novelty here is that, by exploiting the interplay with $U(1)$ anomalies, we can do so simply by integrating an anomaly polynomial, following the arguments set out in §3.1. We thereby *derive* the conditions for global anomaly cancellation in the $\mathbb{Z}/2$ theory from local anomaly cancellation in $U(1)$. We now turn to this calculation.

**Computing the $\eta$-invariant via anomaly interplay**

One choice for the generator of $\Omega_3^{\mathrm{Spin}}(B\mathbb{Z}/2) \cong \mathbb{Z}/8$ is the manifold $\mathbb{R}P^3$ equipped with the nontrivial $\mathbb{Z}/2$ bundle (for which the $\mathbb{Z}/2$-valued holonomy equals $-1$ for the non-trivial homotopy class of loop in $\mathbb{R}P^3 \cong SO(3)$). However, there is another choice of generator for this bordism group that is more convenient to work with in the current context, which can be constructed as a mapping torus for the original 2d theory as follows.[17] Consider first a 2-torus $T^2$ for which $\theta \sim \theta + 2\pi$ and $\chi \sim \chi + 2\pi$ are local 'coordinates', with the spin structure corresponding to antiperiodic boundary conditions chosen in both directions, and where $T^2$ is equipped with a nontrivial $\mathbb{Z}/2$ background gauge field with holonomy minus one around cycles that wrap the $\theta$ coordinate, and holonomy zero otherwise.[18] Now consider a cylinder $T^2 \times [0, 1]$, with $t$ a coordinate on the unit interval, and glue together its two ends to form a three-dimensional mapping torus $M_3$ by identifying

$$(\theta, \chi, 1) \sim (\theta - 2\chi, \chi, 0), \tag{4.9}$$

as well as imposing the anti-periodic spin structure along the new cycle parametrized by $t$.[19] One cannot realise this mapping torus as the boundary of a 4-manifold with both the $\mathbb{Z}/2$ gauge bundle and the spin structure extended; in other words, this mapping torus is in a nontrivial bordism class of $\Omega_3^{\mathrm{Spin}}(B\mathbb{Z}/2) \cong \mathbb{Z}/8$. In fact, it may be taken as a generator of the $\mathbb{Z}/8$.

If one embeds the $\mathbb{Z}/2$ background inside a flat $U(1)$ connection, then one must be able to extend all structures to a 4-manifold bounded by the mapping torus $M_3$ because $\Omega_3^{\mathrm{Spin}}(BU(1)) = 0$. Moreover, computing the exponentiated $\eta$-invariant associated with such a $U(1)$ connection must equal that for the original $\mathbb{Z}/2$ background, by pullback. This will ultimately allow us to derive the $\mathbb{Z}/8$-valued global anomaly in the discrete gauge theory from the local $U(1)$ anomaly, which can be evaluated using the APS index theorem (and hence by simply integrating differential forms over the bounding 4-manifold).[20]

Our task is thus to embed the $\mathbb{Z}/2$ connection inside $U(1)$, and extend both the $U(1)$ connection and the spin structure from the mapping torus described above to a 4-manifold that it bounds. An appropriate flat $U(1)$ connection on $M_3$ with nontrivial holonomy (equal to $-1$) only along cycles that wrap the $\theta$ direction is simply

$$A(\theta, \chi, t) = \frac{1}{2}\mathrm{d}\theta. \tag{4.10}$$

We now extend the mapping torus to a 4-manifold $X_4$ by filling in a 2-disk $D^2$, with radial coordinate $r \in [0, 1]$, that is bounded by the cycle wrapping the $\theta$ coordinate. A suitable extension of the $U(1)$ connection to $X_4$ is then

$$A(r, \theta, \chi, t) = \frac{1}{2}r\mathrm{d}\theta + (1 - r)t\mathrm{d}\chi. \tag{4.11}$$

Note that one cannot simply extend the connection as $A(r, \theta, \chi, t) = r\mathrm{d}\theta/2$ to the whole of $X_4$ because it is incompatible with the construction of the mapping torus, since

---

[17]We thank Philip Boyle Smith for a helpful discussion concerning the construction of this mapping torus.

[18]Equivalently, we consider a "$\mathbb{Z}/2$ defect" along cycles which wrap the $\chi$ coordinate.

[19]This mapping torus is not homeomorphic to a 3-torus, as can be seen by computing the homology groups $H_1(M_3; \mathbb{Z}) \cong \mathbb{Z}^2 \times \mathbb{Z}/2$ and $H_2(M_3; \mathbb{Z}) \cong \mathbb{Z}^2$. We thank Philip Boyle Smith for sharing this computation with us.

[20]If we had instead chosen the manifold $\mathbb{R}P^3$ with non-trivial $\mathbb{Z}/2$ bundle to represent the generator of $\Omega_3^{\mathrm{Spin}}(B\mathbb{Z}/2) \cong \mathbb{Z}/8$, one must be able to extend all structures to a 4-manifold it bounds in a similar fashion by embedding $\mathbb{Z}/2 \subset U(1)$. Indeed, the construction of such a nullbordism is here known, with $\mathbb{R}P^3$ being the asymptotic infinity of the simplest type of asymptotically locally Eucliden (ALE) space [39–41], which is topologically a disk bundle over $S^2$. The integral of the anomaly polynomial on this 4-manifold was computed in Ref. [42], and agrees with our computation of the exponentiated $\eta$-invariant on the 'mapping torus' $M_3$ (see Eq. (4.12)), as it must by bordism invariance. We thank the anonymous journal referee for bringing these results to our attention.

$A(r, \theta, \chi, 1) = r\,\mathrm{d}\theta/2$ is not gauge equivalent (in the bulk) to $A(r, \theta-2\chi, \chi, 0) = (r/2)\mathrm{d}\theta - r\,\mathrm{d}\chi$. (Using Eq. (4.11), we have $A(r, \theta, \chi, 1) = r(\mathrm{d}\theta/2 - \mathrm{d}\chi) + \mathrm{d}\chi \sim A(r, \theta-2\chi, \chi, 0)$, which does respect the gluing condition.)

With this $U(1)$ connection on $X_4$ in hand, which reduces to the desired connection on the $M_3$ boundary, we are now in a position to evaluate the exponentiated $\eta$-invariant $\exp 2\pi i \eta(M_3)$. Firstly, it is straightforward to show that

$$\frac{1}{8\pi^2} \int_{X_4} F \wedge F = \frac{1}{8\pi^2} \int_{X_4} (1-r)\mathrm{d}r \wedge \mathrm{d}\theta \wedge \mathrm{d}t \wedge \mathrm{d}\chi = \frac{1}{4}. \qquad (4.12)$$

Choosing a single Weyl fermion with odd charge under $U(1)$, corresponding to a fermion charged under the original $\mathbb{Z}/2$ (we assume an uncharged fermion of opposite handedness cancels the gravitational anomaly discussed above), the anomaly polynomial is $\Phi_4 = F \wedge F/8\pi^2$. The APS index theorem therefore gives

$$\exp\left(2\pi i \eta(M_3)\right) = \exp\left(2\pi i \frac{1}{8\pi^2} \int_{X_4} F \wedge F\right) = \exp\left(\frac{2\pi i}{4}\right), \qquad (4.13)$$

and so the original $\mathbb{Z}/2$-charged single Weyl fermion is associated with a mod 4 global anomaly.

Going back to our anomaly pullback map $\pi^*$, we learn that the $\mathbb{Z}/2$-charged Weyl fermion has an anomaly

$$\nu_{\mathrm{Weyl}} = 2 \bmod 8, \qquad (4.14)$$

and so the anomaly pullback map $\pi^*$ is given by

$$\pi^* : H^4_{I\mathbb{Z}}(MT(\mathrm{Spin} \times U(1))) \cong \mathbb{Z} \times \mathbb{Z} \to H^4_{I\mathbb{Z}}(MT(\mathrm{Spin} \times \mathbb{Z}/2)) \cong \mathbb{Z}/8 \times \mathbb{Z} :$$
$$\left(\mathcal{A}_{\mathrm{gauge}}, \mathcal{A}_{\mathrm{grav}}\right) \mapsto \left(2\mathcal{A}_{\mathrm{gauge}} \bmod 8, \mathcal{A}_{\mathrm{grav}}\right). \qquad (4.15)$$

In contrast to all the other examples we discuss below, the pullback is not surjective. This is simply because a *single* $\mathbb{Z}/2$-charged Majorana–Weyl fermion, which is known to generate the mod 8 anomaly, cannot be embedded as a representation of $\mathrm{Spin}(2) \times U(1)$; rather, we need at least of pair of such Majorana–Weyls to compose a fundamental Weyl fermion of the $U(1)$ gauge theory. Viewing the anomaly interplay as a tool for calculating the global anomaly in the $\mathbb{Z}/2$ gauge theory, we therefore only learn from our computation that a single Majorana–Weyl fermion contributes an anomaly of

$$\nu_{\mathrm{Maj}} = 1 \text{ or } 5 \bmod 8, \qquad (4.16)$$

and there is no way to decide between these two answers using this particular anomaly interplay.[21] Nonetheless, the conditions for cancelling the global anomaly can still be inferred from this computation, because we learn (in either case) that the anomaly for a single Majorana–Weyl is order 8. Therefore 8 such fermions (of the same chirality) are needed to cancel the anomaly. More generally, the mod 8 anomaly counts the number of left-moving Majorana–Weyl fermions minus the right-moving ones.

This conclusion, which we have derived from a purely local $U(1)$ anomaly using the interplay diagram (4.5), agrees with other examples of $\mathbb{Z}/8$ anomalies in fermionic system with a unitary $\mathbb{Z}/2$ symmetry [43, 45, 46], where a Majorana–Weyl fermion that couples to the $\mathbb{Z}/2$ symmetry contributes the finest anomaly of 1 mod 8.

---

[21]The answer, which corresponds to evaluating the $\eta$-invariant for a single Majorana–Weyl on the mapping torus $M_3$ described above (alternatively, on $\mathbb{R}P^3$ with non-trivial $\mathbb{Z}/2$ bundle, to which $M_3$ is bordant), is $\nu_{\mathrm{Maj}} = 1 \bmod 8$. This can be computed by other means [43], for example by a CFT calculation [44].

# 5  4d anomalies in $U(2)$ *vs.* $SU(2)$

## 5.1  The spin case

For our next example, we move up two dimensions and revisit the anomaly interplay between $U(2)$ and $SU(2)$ gauge theories in $d = 4$, each defined using a spin structure [16] (see also [47]). The map $\pi : SU(2) \to U(2)$ will denote the usual embedding of $SU(2) \subset U(2)$ as the subgroup of 2-by-2 unitary matrices that have determinant one.

**Bordism account of the anomalies**

Similar to before, we need to compute the torsion subgroup of $\Omega_5$ and the free part of $\Omega_6$, for each $G$, in order to compute the short exact sequences (2.5). Computations using spectral sequences (see Refs. [23, 47, 48]) yield

$$\Omega_5^{\mathrm{Spin}}(BSU(2)) \cong \mathbb{Z}/2, \qquad \Omega_6^{\mathrm{Spin}}(BSU(2)) \cong \mathbb{Z}/2, \tag{5.1}$$

for $SU(2)$, and

$$\Omega_5^{\mathrm{Spin}}(BU(2)) = 0, \qquad \Omega_6^{\mathrm{Spin}}(BU(2)) \cong \mathbb{Z}^3, \tag{5.2}$$

for $U(2)$. The short exact sequences are thus, with $SU(2)$ along the top row and $U(2)$ along the bottom row, simply

$$
\begin{array}{ccccccccc}
0 & \longrightarrow & \mathbb{Z}/2 & \longrightarrow & \mathbb{Z}/2 & \longrightarrow & 0 & \longrightarrow & 0 \\
 & & \Big\uparrow & & {\pi^*}\Big\uparrow & & \Big\uparrow & & \\
0 & \longrightarrow & 0 & \longrightarrow & \mathbb{Z}^3 & \longrightarrow & \mathbb{Z}^3 & \longrightarrow & 0.
\end{array}
\tag{5.3}
$$

For $SU(2)$ the vanishing of the right factor accords with there being no local anomaly, but there is the $\mathbb{Z}/2$-valued global anomaly discovered by Witten [19]. This global anomaly cancels if and only if there is an even number of fermions with $SU(2)$ isospins in the set $2\mathbb{Z}_{\geq 0} + 1/2$.

For $G = U(2)$, the vanishing of $\Omega_5$ means that there is no global anomaly, even though $\pi_4(U(2)) = \mathbb{Z}/2$, associated with a homotopically non-trivial large gauge transformation in the $SU(2)$ subgroup.[22] However, there are local anomalies when $G = U(2)$, and we next describe explicitly how these are also detected (up to deformation class) by bordism, only now it is the free part of the bordism group in degree $d + 2 = 6$.

The local anomaly for $G = U(2)$ is classified by three integers which are linear combinations of the three anomaly coefficients: (i) the cubic $U(1)$ anomaly $\mathcal{A}_{\mathrm{cub}}$, (ii) the mixed $U(1) \times SU(2)^2$ anomaly $\mathcal{A}_{\mathrm{mix}}$, and (iii) the $U(1)$-gravitational anomaly $\mathcal{A}_{\mathrm{grav}}$. Supposing there are $N_j$ fermions transforming in isospin-$j$ representations of $U(2)$, with $U(1)$ charges $\{q_{j,\alpha}\} \equiv 2j \pmod{2}$[23] for $\alpha = 1, \dots, N_j$, denoted symbolically by a general $U(2)$ representation $\rho = \bigoplus_j \bigoplus_{\alpha=1}^{N_j} (\mathbf{2j+1}, q_{j,\alpha})$, these anomaly coefficients can be written

$$
\begin{aligned}
\mathcal{A}_{\mathrm{cub}} &= \sum_j (2j + 1) \sum_\alpha (q_{j,\alpha})^3, \\
\mathcal{A}_{\mathrm{mix}} &= \sum_j T(j) \sum_\alpha q_{j,\alpha}, \\
\mathcal{A}_{\mathrm{grav}} &= \sum_j (2j + 1) \sum_\alpha q_{j,\alpha},
\end{aligned}
\tag{5.4}
$$

---

[22]This is another example that shows how naïve homotopy-based reasoning can swiftly lead to incorrect conclusions regarding global anomalies [30].

[23]This 'isospin-charge relation' that links the $U(1)$ charge to the $SU(2)$ isospin is a consequence of the $\mathbb{Z}/2$ quotient in the definition of $U(2) \equiv (SU(2) \times U(1))/\mathbb{Z}/2$.

where $T(j) = \frac{2}{3}j(j+1)(2j+1)$ is the $SU(2)$ Dynkin index.

To find a basis of linear combinations of these three anomaly coefficients that can be used as labels for the $U(2)$ local anomaly, we need to find a basis for $\mathrm{Hom}(\Omega_6^{\mathrm{Spin}}(BU(2)),\mathbb{Z})$ dual to the bordism group $\Omega_6^{\mathrm{Spin}}(BU(2))$. In terms of the first Pontryagin class $p_1$ of the tangent bundle and the Chern classes $c_1, c_2$ of the $U(2)$ bundle, we can choose our basis to be

$$\alpha_1 = c_1^3, \quad \alpha_2 = -\frac{1}{2}c_1c_2, \quad \alpha_3 = \frac{1}{6}\left(c_1^3 - \frac{1}{4}p_1c_1\right). \tag{5.5}$$

To see this, first note that $\mathrm{Hom}(\Omega_6^{\mathrm{Spin}}(BU(2)),\mathbb{Z}) \otimes \mathbb{Q}$ is isomorphic to $H^6(B\mathrm{Spin} \times BU(2);\mathbb{Q})$, which is generated by $p_1 \cup ch_1$ (henceforth just $p_1 ch_1$), $ch_1 \cup ch_1 \cup ch_1$ (henceforth just $ch_1^3$), and $ch_3$, where $p_1$ is the first Pontryagin class and $ch_1, ch_3$ are the first and third Chern characters of the $U(2)$ bundle. So a basis for $\mathrm{Hom}(\Omega_6^{\mathrm{Spin}}(BU(2)),\mathbb{Z})$ must consist of three independent rational linear combinations of $p_1 ch_1$, $ch_1^3$, and $ch_3$ that are integral. They can be found by evaluating the anomaly polynomial, which is integral by virtue of the Atiyah–Singer index theorem, on various $U(2)$ representations. Our choice of basis above arises from considering the anomaly polynomial with representations $(\mathbf{1}, 2)$, $(\mathbf{1}, 4)$ and $(\mathbf{2}, 1)$, and expressing the Chern characters in terms of Chern classes.

One set of generators of the bordism group $\Omega_6^{\mathrm{Spin}}(BU(2))$ that are dual to the basis $\{\alpha_1, \alpha_2, \alpha_3\}$ are as follows. Firstly, a manifold dual to $\alpha_1$ is $(\mathbb{C}P^3, c)$, the complex projective 3-space equipped with a $U(2)$ bundle whose first Chern class $c_1$ is given by the canonical generator $c \in H^2(\mathbb{C}P^3;\mathbb{Z})$, and whose second Chern class $c_2 = 0$. (This corresponds to embedding the canonical complex line bundle over $\mathbb{C}P^3$ inside a $U(2)$ bundle.) Secondly, a manifold dual to $\alpha_2$ is the direct product manifold $\mathbb{H}P^1 \times \mathbb{C}P^1 \cong S^4 \times S^2$, equipped with a $U(2)$ bundle corresponding to a 1-instanton on the $S^4$ and a 2-monopole on the $S^2$. More precisely, the bundle has $c_2[S^4] = -1$ and $c_1[S^2] = 2$. We denote this generator by $S_1^4 \times S_2^2$. Lastly, a manifold dual to $\alpha_3$ is $\mathbb{C}P^1 \times \mathbb{C}P^1 \times \mathbb{C}P^1 \cong S^2 \times S^2 \times S^2$, equipped with a $U(2)$ bundle corresponding to a 1-monopole on each factor, that is, the first Chern class is given by $c_1 = a_1 + a_2 + a_3$ where $a_i \in H^2(\mathbb{C}P^1;\mathbb{Z})$ is the canonical generator of the second integral cohomology group of each 2-sphere factor. We denote this generator by $S_1^2 \times S_1^2 \times S_1^2$. The dual pairing between these three generators and $\alpha_1, \alpha_2, \alpha_3$ is given by

$$\begin{aligned}
\alpha_1[(\mathbb{C}P^3, c)] &= 1, & \alpha_1[S_1^4 \times S_2^2] &= 0, & \alpha_1[S_1^2 \times S_1^2 \times S_1^2] &= 6, \\
\alpha_2[(\mathbb{C}P^3, c)] &= 0, & \alpha_2[S_1^4 \times S_2^2] &= 1, & \alpha_2[S_1^2 \times S_1^2 \times S_1^2] &= 0, \\
\alpha_3[(\mathbb{C}P^3, c)] &= 0, & \alpha_3[S_1^4 \times S_2^2] &= 0, & \alpha_3[S_1^2 \times S_1^2 \times S_1^2] &= 1.
\end{aligned} \tag{5.6}$$

A general element of the group $\mathrm{Hom}(\Omega_6^{\mathrm{Spin}}(BU(2)),\mathbb{Z})$ can then be recast in terms of the more familiar anomaly coefficients (5.4) by integrating the (appropriately normalized) anomaly polynomial $\Phi_6 = \hat{A}\,\mathrm{tr}_\rho \exp(\mathcal{F}/2\pi)|_6$, where $\hat{A}$ is the Dirac genus of the tangent bundle, $\mathcal{F}$ is the $U(2)$ field strength, and $\rho$ is the representation specified above. By expanding $\mathcal{F}$ in terms of the $U(1)$ field strength $f$ and $SU(2)$ field strength $F$ via[24]

$$\mathcal{F} = \frac{f}{2}\mathbf{1}_2 + F, \tag{5.7}$$

and expanding $\hat{A}$ in terms of the first Pontryagin class $p_1$, we can express the anomaly polynomial as

$$\Phi_6 = \frac{1}{3!}\mathrm{tr}_\rho\left(\frac{\mathcal{F}}{2\pi}\right)^3 - \frac{1}{24}p_1\mathrm{tr}_\rho\frac{\mathcal{F}}{2\pi} = \mathcal{A}_{\mathrm{cub}}\frac{1}{48}\left(\frac{f}{2\pi}\right)^3 + \frac{\mathcal{A}_{\mathrm{mix}}}{4}\frac{f}{2\pi}\mathrm{tr}\left(\frac{F}{2\pi}\right)^2 - \frac{\mathcal{A}_{\mathrm{grav}}}{48}p_1\frac{f}{2\pi}, \tag{5.8}$$

---

[24]We use the convention that $f/2\pi$ represents the first Chern class $c_1$ of $U(2)$.

where the anomaly coefficients are given in Eq. (5.4). By writing $f/2\pi = c_1$ and $\operatorname{tr}(F/2\pi)^2 = -2c_2 + c_1^2/2$, the anomaly polynomial for the representation $\rho$ can be expressed in terms of the basis $\{\alpha_1, \alpha_2, \alpha_3\}$ as

$$\Phi_6 = \frac{\mathcal{A}_{\text{cub}} - 4\mathcal{A}_{\text{grav}} + 6\mathcal{A}_{\text{mix}}}{48}\alpha_1 + \mathcal{A}_{\text{mix}}\alpha_2 + \frac{\mathcal{A}_{\text{grav}}}{2}\alpha_3. \tag{5.9}$$

Thus, $\operatorname{Hom}(\Omega_6^{\text{Spin}}(BU(2)), \mathbb{Z})$ should be identified with

$$\mathbb{Z}\alpha_1 \oplus \mathbb{Z}\alpha_2 \oplus \mathbb{Z}\alpha_3,$$

an element of which is labelled by the following three linear combinations of the anomaly coefficients,

$$(r, s, t) = \left(\frac{1}{48}\left(\mathcal{A}_{\text{cub}} - 4\mathcal{A}_{\text{grav}} + 6\mathcal{A}_{\text{mix}}\right), \mathcal{A}_{\text{mix}}, \frac{1}{2}\mathcal{A}_{\text{grav}}\right). \tag{5.10}$$

For example, a fermion transforming under $U(2)$ as an $SU(2)$ singlet with $U(1)$ charge $q = 2$ corresponds to a local anomaly in the deformation class $(0, 0, 1) \in H_{I\mathbb{Z}}^6(MT(\text{Spin} \times U(2))) \cong \mathbb{Z}^3$.

**Deducing the anomaly interplay maps**

The commutative diagram (5.3) will encode a non-trivial anomaly interplay if $\pi^*$ is not the zero map, meaning that a local $U(2)$ anomaly can pullback to a global anomaly in $SU(2)$. To see that this is the case, it suffices to consider a fermion $\psi$ transforming in the $U(2)$ representation

$$\psi \sim (\mathbf{2}, q), \qquad q \in 2\mathbb{Z} + 1, \tag{5.11}$$

*i.e.* as a doublet under the $SU(2)$ subgroup. The vanishing of $\Omega_5^{\text{Spin}}(BU(2))$ implies that any closed 5-manifold $X$ equipped with spin structure and a $U(2)$ gauge bundle is the boundary of a 6-manifold $Y$ to which these structures extend. Hence on $X = \partial Y$ the anomaly theory for this representation evaluates to $\exp\left(2\pi i \int_Y \Phi_6\right)$ by the APS index theorem.

The $U(2)$ representation (5.11) is associated with a trio of local anomalies, corresponding to the element $((q^3 - q)/24, q, q) \in \operatorname{Hom}(\Omega_6^{\text{Spin}}(BU(2)), \mathbb{Z}) \cong H_{I\mathbb{Z}}^6(MT(\text{Spin} \times U(2)))$. To pullback this element to $H_{I\mathbb{Z}}^6(MT(\text{Spin} \times SU(2))) \cong \mathbb{Z}/2$ along $\pi^*$, we decompose (5.11) into representations of the subgroup, here simply $(\mathbf{2}, q) \to \mathbf{2}$, and evaluate the anomaly theory for a single $SU(2)$ doublet on an arbitrary closed 5-manifold. Because the free part of the $6^{\text{th}}$ bordism group of $SU(2)$ vanishes (*i.e.* because there are no local anomalies), it is sufficient to evaluate the exponentiated $\eta$-invariant on the generator of the bordism group $\Omega_5^{\text{Spin}}(BSU(2))$. We know that a suitable generator of $\Omega_5^{\text{Spin}}(BSU(2))$ is a mapping torus $S^4 \times_g S^1$ that is glued together using a homotopically non-trivial gauge transformation $g(x) \in \pi_4(SU(2))$, on which $\exp\left(2\pi i \eta_{S^4 \times_g S^1}\right) = -1$ for the doublet representation. (This equals the phase accrued by the fermionic partition function under this gauge transformation, as originally computed by Witten using spectral flow [19].) Thus, we know that the pullback map $\pi^*$ between anomaly theories is such that $\mathbb{Z}^3 \ni ((q^3 - q)/24, q, q) \mapsto 1 \in \mathbb{Z}/2$, which already tells us that $\pi^*$ is a surjection. Performing a similar calculation for an arbitrary $U(2)$ fermion representation, one can deduce that the anomaly pullback map is

$$\pi^* : H_{I\mathbb{Z}}^6(MT(\text{Spin} \times U(2))) \cong \mathbb{Z}^3 \to H_{I\mathbb{Z}}^6(MT(\text{Spin} \times SU(2))) \cong \mathbb{Z}/2 :$$

$$\left(\frac{1}{48}\left(\mathcal{A}_{\text{cub}} - 4\mathcal{A}_{\text{grav}} + 6\mathcal{A}_{\text{mix}}\right), \mathcal{A}_{\text{mix}}, \frac{1}{2}\mathcal{A}_{\text{grav}}\right) \mapsto \mathcal{A}_{\text{mix}} \bmod 2. \tag{5.12}$$

As a result, the splitting maps in (5.3) do not give rise to commuting squares, on either the left or right, which is the mathematical statement of this anomaly interplay.

**Reversing the logic: using interplay to derive the $SU(2)$ anomaly**

In the account just given, we deduced what the anomaly pullback map was by first decomposing representations of the locally-anomalous theory into representations of $SU(2)$, then invoking a known result for $\exp 2\pi i\eta$ evaluated on a 5-manifold in the non-trivial bordism class of $\Omega_5^{\mathrm{Spin}}(BSU(2)) \cong \mathbb{Z}/2$. Now, as demonstrated in §4 for a 2d example, one can in fact turn the argument around to *derive* the conditions for global anomaly cancellation in the $SU(2)$ theory from local anomaly cancellation in $U(2)$.

The first step is to view a representative of the generator of $\Omega_5^{\mathrm{Spin}}(BSU(2)) \cong \mathbb{Z}/2$ as a manifold with a Spin $\times U(2)$ structure by embedding the $SU(2)$ connection inside $U(2)$. The result must be nullbordant as a $U(2)$ bundle because $\Omega_5^{\mathrm{Spin}}(BU(2)) = 0$. One may choose this representative to be the original mapping torus $S^4 \times_g S^1$ of Witten, described above. But due to bordism invariance, we are free to make alternative, simpler choices of representative in the same bordism class. In particular, we consider a mapping torus $X = S^4 \times S^1$, equipped with an $SU(2)$-connection $A_{\mathrm{inst}}$ with instanton number one through the $S^4$ factor, and a periodic spin structure along $S^1$ [34]. This choice of spin structure corresponds to implementing a transformation by $(-1)^F$, or equivalently a constant gauge transformation by $-\mathbf{1} \in SU(2)$, upon going once round the mapping torus. To appreciate why this manifold is not nullbordant, it is perhaps enlightening to at least explain why naïve extensions, in which we fill in either the $S^1$ or the $S^4$ with a 2- or 5-disk, do not succeed; the former cannot be done because the periodic spin structure corresponds to the non-trivial class in $\Omega_1^{\mathrm{Spin}}(\mathrm{pt}) \cong \mathbb{Z}/2$, while the latter cannot be done due to the non-zero instanton number threading the $S^4$.[25]

This discussion also makes it clear why, when considered as a $U(2)$ bundle, both the gauge bundle and spin structure can be extended straightforwardly to a 6-manifold bounded by this mapping torus. To see this, we first observe that the twist by $(-1)^F$ can be encoded in the $U(2)$ connection, via

$$A_\phi = \frac{1}{2}\mathrm{d}\phi\,\mathbf{1}_2 + A_{\mathrm{inst}}\,, \tag{5.13}$$

where the coordinate $\phi \in [0, 2\pi)$ parametrizes the $S^1$ direction, and $A_{\mathrm{inst}}$ denotes the $SU(2)$ 1-instanton field configuration on $S^4$. Crucially, this allows us to take the anti-periodic spin structure around the $S^1$ factor, corresponding now to the trivial class in $\Omega_1^{\mathrm{Spin}}(\mathrm{pt})$. Moreover, the $U(2)$ gauge bundle can also be extended to a six-manifold $Y = S^4 \times D^2$, where $D^2$ denotes a hemisphere bounded by the $S^1$ factor in $X$ (so that $\partial Y = S^4 \times S^1$), by taking the $U(1)$ component $f$ of the $U(2)$ field strength as defined in (5.7) to be that of a magnetic monopole with charge $g = 2$ placed at the centre of the hemisphere (see Fig. 1 for a cartoon illustration). More precisely, we take $D^2$ to be one half of a 2-sphere $S^2$ with $c_1[S^2] = \int_{S^2} f/(2\pi) = 2$. This completes the construction of an explicit nullbordism for the $SU(2)$ mapping torus by embedding $SU(2) \subset U(2)$.

With a suitable nullbordism $Y$ in hand, we can then evaluate the anomaly theory for the $(\mathbf{2}, q)$ representation using the APS index theorem, by which our task reduces to the integration of a differential form,

$$\exp\left(2\pi i\eta(M \times S^1)\right) = \exp\left(2\pi iq \int_{D^2} \frac{1}{2}\frac{f}{2\pi} \int_M \frac{1}{8\pi^2}\mathrm{tr}\,F \wedge F\right) = \exp(i\pi q) = -1\,, \tag{5.14}$$

where we used the fact that the monopole is spherically symmetric to write $\int_{D^2} f/(2\pi) = c_1[S^2]/2$ in the penultimate step, and that $q$ is necessarily odd in the last step.

---

[25]These particular 'obstructions' are encoded in the second page of the Atiyah–Hirzebruch spectral sequence used to compute $\Omega_d^{\mathrm{Spin}}(BSU(2))$; the stabilisation of this spectral sequence indeed reveals that it is impossible to construct *any* nullbordism for this mapping torus.

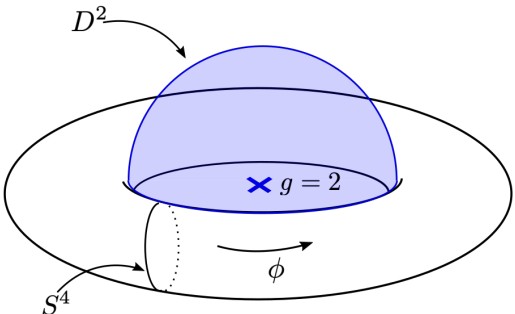

**Figure 1:** Schematic illustration of the extension of the mapping torus $S^4 \times S^1$ to the 6-manifold $S^4 \times D^2$, made possible by embedding $SU(2)$ as a subgroup of $U(2)$.

By pullback, this must equal the evaluation of the $SU(2)$ anomaly theory for the doublet representation on the original mapping torus, which therefore suffers from a $\mathbb{Z}/2$-valued global anomaly. By repeating the exercise for other representations, one learns that in general the anomaly pullback map $\pi^*$ is given by (5.12). Hence, the $SU(2)$ anomaly vanishes if and only if

$$\mathcal{A}_{\text{mix}} = \sum_j T(j) \sum_\alpha q_{j,\alpha} = 0 \mod 2\,, \tag{5.15}$$

for the $U(2)$ local anomaly. Now, $T(j)$ is odd if and only $j \in 2\mathbb{Z}_{\geq 0} + 1/2$, and for these half-integral isospins $q$ must be odd, so this equation becomes [16]

$$\sum_{j \in 2\mathbb{Z}_{\geq 0}+1/2} 1 = 0 \mod 2\,, \tag{5.16}$$

which, upon decomposing to irreps of $SU(2)$, is precisely the condition for $SU(2)$ anomaly cancellation. This amounts to a rigorous derivation of the $SU(2)$ anomaly constraint using the APS index theorem for nullbordant manifolds in $U(2)$, which is arguably easier than computing $\eta$ on non-trivial bordism classes in the original theory. Because the most general anomaly in the $SU(2)$ theory is classified by $H^6_{I\mathbb{Z}}(MT(\text{Spin} \times SU(2))) \cong \mathbb{Z}/2$, there can be no further conditions for $SU(2)$ anomaly cancellation.

## 5.2 Non-spin generalization: the '$w_2 w_3$ anomaly'

In Ref. [34] it was shown that if one instead defines a 4d $SU(2)$ gauge theory by extending the spacetime symmetry group Spin(4) non-trivially by $SU(2)$, then there is a new $\mathbb{Z}/2$-valued global anomaly for fermions in the isospin representations $4r + 3/2$, $r \in \mathbb{Z}_{\geq 0}$. In particular, the symmetry structure is

$$H = \frac{\text{Spin} \times SU(2)}{\mathbb{Z}/2}\,,$$

in which the $\mathbb{Z}/2$ quotient identifies the central element $-\mathbf{1} \in SU(2)$ with $(-1)^F$ in Spin, which we abbreviate to $H = \text{Spin-}SU(2)$.

The relevant bordism group capturing global anomalies (there are no local anomalies) is

$$\Omega_5^{\text{Spin-}SU(2)} \cong \mathbb{Z}/2 \times \mathbb{Z}/2\,. \tag{5.17}$$

A suitable generator for the first $\mathbb{Z}/2$ factor is given by the Dold manifold $D = (\mathbb{C}P^2 \times S^1)/(\mathbb{Z}/2)$ (where the $\mathbb{Z}/2$ quotient identifies complex conjugation on $\mathbb{C}P^2$ with the antipodal map on

the circle), equipped with a certain Spin-$SU(2)$ structure. This provides a mapping torus on which one can detect the 'new $SU(2)$ anomaly' of Ref. [34], where the mapping torus is glued using a combined diffeomorphism and gauge transformation. A bordism invariant that evaluates to 1 mod 2 on the Dold manifold $D$ is provided by $w_2 \cup w_3$ (henceforth just denoted $w_2 w_3$), the cup product of the second and third Stiefel–Whitney classes of the tangent bundle, sometimes referred to as the 'de Rham invariant'. This bordism invariant can be obtained as the exponentiated $\eta$-invariant for a single fermion in the isospin $j$ representation of $SU(2)$ for any $j \in 4\mathbb{Z}_{\geq 0} + 3/2$.

The second $\mathbb{Z}/2$ factor in the bordism group (5.17), which is generated by the mapping torus $X = S^4 \times S^1$ with $SU(2)$ instanton number one through the $S^4$ factor, captures the original $SU(2)$ anomaly of Witten. As we have seen above, a bordism invariant that evaluates to 1 mod 2 on $X$ is the exponentiated $\eta$-invariant for a fermion in the doublet representation of $SU(2)$.

In analogy with the anomaly interplay between Spin $\times SU(2)$ and Spin $\times U(2)$ studied in the previous Section, we here embed the symmetry type $H = $ Spin-$SU(2)$ in the symmetry type $H' = $ Spin-$U(2)$, whose relevant bordism groups are given by

$$\Omega_5^{\text{Spin-}U(2)} \cong \mathbb{Z}/2, \qquad \Omega_6^{\text{Spin-}U(2)} \cong \mathbb{Z} \times \mathbb{Z} \times \mathbb{Z}. \tag{5.18}$$

(See Appendix A.2 for the computation of the sixth bordism group here.) The $\mathbb{Z}/2$ factor in $\Omega_5^{\text{Spin-}U(2)}$ is again generated by the Dold manifold $D$, now with a certain Spin-$U(2)$ structure. The embedding $H \to H'$ induces the anomaly interplay diagram

$$\begin{array}{ccccccccc}
0 & \longrightarrow & \mathbb{Z}/2 \times \mathbb{Z}/2 & \longrightarrow & \mathbb{Z}/2 \times \mathbb{Z}/2 & \longrightarrow & 0 & \longrightarrow & 0 \\
 & & \big\uparrow & & {\scriptstyle\pi^*}\big\uparrow & & \big\uparrow & & \\
0 & \longrightarrow & \mathbb{Z}/2 & \longrightarrow & \mathbb{Z}/2 \times \mathbb{Z}^3 & \longrightarrow & \mathbb{Z}^3 & \longrightarrow & 0.
\end{array} \tag{5.19}$$

The pullback $\pi^*$ maps the local anomaly factor $\mathbb{Z}^3$ into the second $\mathbb{Z}/2$ factor of $\Omega_5^{\text{Spin-}SU(2)}$ exactly as in the interplay between $SU(2)$ and $U(2)$.

The role of the first $\mathbb{Z}/2$ factor is in a sense more subtle, but in another sense rather trivial. While the same Dold manifold described above can be taken as a generator of the $\mathbb{Z}/2$ factor in the bordism group, the corresponding element in the *co*bordism group can in fact never be realised as a fermionic global anomaly in the Spin-$U(2)$ theory. This can be seen from the fact that the conditions for local anomaly cancellation (including only spin-1/2 fermions) in the Spin-$U(2)$ theory preclude a non-vanishing 'new $SU(2)$ anomaly', as shown in Ref. [16]. In other words, evaluating the exponentiated $\eta$-invariant for any set of chiral spin-1/2 fermions coupled to the Spin-$U(2)$ connection can never give a non-trivial phase on $D$ when the local anomalies cancel. This is the first concrete example we have seen of a non-trivial invertible phase that appears in the cobordism group, but which cannot be realised as an anomaly due to spin-1/2 chiral fermions.

Despite this, it is possible to couple the Spin-$U(2)$ theory directly to a topological quantum field theory (TQFT) whose partition function is given by $(-1)^{w_2 w_3}$ on closed 5-manifolds, thereby generating an invertible phase in the non-trivial element of $\mathbb{Z}/2 \subset H_{I\mathbb{Z}}^6(MT(\text{Spin-}U(2)))$ which reproduces the same anomaly. (In a sense this is a 'pure-gravitational' anomaly, being computed from characteristic classes of the tangent bundle.) The anomaly pullback map $\pi^*$ acts as the identity map on the first $\mathbb{Z}/2$ factor, mapping the cohomology class $w_2 w_3$ to itself. The crucial difference is that in the Spin-$SU(2)$ theory (unlike for Spin-$U(2)$), the anomaly theory $(-1)^{w_2 w_3}$ can be reproduced by the exponentiated $\eta$-invariant, say for a single isospin-3/2 fermion coupled to a particular Spin-$SU(2)$ connection.

### 5.3 A remark concerning the 5d $SU(2)$ anomaly

The fact that $\Omega_6^{\text{Spin}}(BSU(2)) \cong \mathbb{Z}/2$ corresponds to a $\mathbb{Z}/2$-valued global anomaly in $SU(2)$ gauge theory in 5d, which is generated by a 'symplectic Majorana fermion' multiplet [49]. The bordism group $\Omega_6(BU(2))$, however, is torsion-free, ruling out a global anomaly in the $U(2)$ version. In analogy with the 4d case, one might again wonder what has become of the 5d $SU(2)$ global anomaly when $SU(2)$ is embedded as a subgroup in $U(2)$, and whether there is a similar interplay with local anomalies. The story here is in fact much more mundane; $\Omega_7(BU(2))$ vanishes, so there are no anomalies whatsoever in a 5d $U(2)$ gauge theory. The resolution to this little puzzle is simply that one cannot embed the symplectic Majorana multiplet, responsible for the 5d $SU(2)$ global anomaly, into representations of $U(2)$; the isospin-charge relation in $U(2)$ means any $SU(2)$ doublet must have non-zero $U(1)$ charge, and so cannot be Majorana.

# 6 4d $SU(2)$ anomaly from $SU(3)$

Elitzur and Nair, inspired by the emergence of the $SU(2)$ global anomaly from the $SU(3)$ WZW term in the chiral lagrangian [18], showed how the $SU(2)$ global anomaly associated with the doublet representation can be computed from a local anomaly not in $U(2)$, but in $SU(3)$ [17]. To our knowledge, this was the first instance in which a local anomaly was essentially 'pulled back' to derive a global anomaly, implicitly exploiting the possible interplay between the two. Their method was based on the homotopy groups of $SU(2)$ and $SU(3)$, which fit inside a long exact sequence together with the homotopy groups of $SU(3)/SU(2) \cong S^5$, and embedding $SU(2)$ gauge field configurations inside $SU(3)$.

The bordism-based formalism which we use here is significantly more powerful. Because $SU(2)$ has no local anomalies in 4d, bordism invariance allows one to analyse the most general possible global anomaly theory by performing only a handful of computations, accounting for all possible $SU(2)$ bundles over all possible mapping tori. Moreover because the local anomaly for $SU(3)$ has a single generator, we will in fact only need to do a single computation, namely 'pushing forward' the $SU(2)$ single instanton mapping torus and then evaluating the $SU(3)$ local anomaly for the triplet using the APS index theorem. This will be enough to recover complete information about the $SU(2)$ global anomaly for arbitrary representations on arbitrary closed manifolds.

To deduce the anomaly interplay map for the subgroup embedding $\pi : SU(2) \to SU(3)$ (in which $SU(2)$ is embedded in, say, the upper-left 2-by-2 block of $SU(3)$), we need the bordism groups for $BSU(2)$ which were already given above (5.1), and we need

$$\Omega_5^{\text{Spin}}(BSU(3)) = 0, \qquad \Omega_6^{\text{Spin}}(BSU(3)) \cong \mathbb{Z}, \tag{6.1}$$

for $SU(3)$. We then have the following pair of short exact sequences

$$
\begin{array}{ccccccccc}
0 & \longrightarrow & \mathbb{Z}/2 & \longrightarrow & \mathbb{Z}/2 & \longrightarrow & 0 & \longrightarrow & 0 \\
 & & \Big\uparrow & & {\pi^*}\Big\uparrow & & \Big\uparrow & & \\
0 & \longrightarrow & 0 & \longrightarrow & \mathbb{Z} & \longrightarrow & \mathbb{Z} & \longrightarrow & 0 \,.
\end{array}
\tag{6.2}
$$

The top line encodes the $\mathbb{Z}/2$-valued global anomaly in $SU(2)$, and the bottom line encodes the $SU(3)$ perturbative anomaly, a generator for which is a single fermion in the fundamental **3** representation.

In general, the $SU(3)$ anomaly coefficient $\mathcal{A}_{SU(3)}$ is obtained by summing over the contributions from some number $n(a_1, a_2)$ of left-handed fermion multiplets in each $SU(3)$ irreducible

representation, indicated by the Dynkin labels $(a_1, a_2)$, *viz.*

$$\mathcal{A}_{SU(3)} = \sum_{a_1, a_2 \in \mathbb{N}} n(a_1, a_2) \mathcal{A}(a_1, a_2) \,, \tag{6.3}$$

where the individual anomaly coefficients are (*c.f.* page 72 of [50])

$$\mathcal{A}(a_1, a_2) = \frac{1}{120}(a_1 - a_2)(2a_1 + a_2 + 3)(a_1 + 2a_2 + 3)(a_1 + 1)(a_2 + 1)(a_1 + a_2 + 2) \,. \tag{6.4}$$

(One can verify that, with this normalization, the anomaly coefficient for the fundamental representation $(1, 0)$ equals one.) We now ask whether there is an anomaly interplay, in other words can local anomalies in $SU(3)$ pullback to the non-trivial $SU(2)$ anomaly? To answer this question, it is sufficient to decompose the triplet representation $\mathbf{3} \to \mathbf{2} \oplus \mathbf{1}$ into representations of $SU(2)$ and compute the anomaly theory. Since there is an unpaired $SU(2)$ doublet (and an irrelevant $SU(2)$ singlet) this theory has a global $SU(2)$ anomaly. This is enough to completely fix the anomaly pullback map $\pi^*$ to be the non-trivial homomorphism from $\mathbb{Z}$ to $\mathbb{Z}/2$, thus

$$\pi^* : H^6_{I\mathbb{Z}}(MT(\mathrm{Spin} \times SU(3))) \cong \mathbb{Z} \to H^6_{I\mathbb{Z}}(MT(\mathrm{Spin} \times SU(2))) \cong \mathbb{Z}/2 :$$
$$\mathcal{A}_{SU(3)} \mapsto \mathcal{A}_{SU(3)} \bmod 2 \,. \tag{6.5}$$

Now, we can once again turn this argument on its head, pretending for a moment that we do not know the conditions for anomaly cancellation in $SU(2)$, and we can derive them from the perturbative $SU(3)$ anomaly. The argument would be as follows. Firstly, to compute a global anomaly for the $SU(2)$ doublet representation, one views the mapping torus equipped with a single $SU(2)$ instanton as a manifold with a Spin $\times SU(3)$ structure by embedding the $SU(2)$ connection inside $SU(3)$. This is exactly as in the previous section. This mapping torus, which is nullbordant in $\Omega_5^{\mathrm{Spin}}(BSU(3)) = 0$, can be extended to a bounding 6-manifold $Y$ with $SU(3)$ connection (in fact using precisely the same $U(2)$ connection of the previous section and embedding $U(2)$ in $SU(3)$). Thus, for any $SU(3)$ representation we can evaluate $\exp 2\pi i \eta$ on the pushed forward mapping torus by integrating the $SU(3)$ anomaly polynomial on $Y$. When we do this for the triplet we compute $\exp 2\pi i \eta_{M \times S^1} = -1$, and because $\mathbf{3} \to \mathbf{2} \oplus \mathbf{1}$ under $SU(3) \to SU(2)$ this must coincide with the anomaly theory for the $SU(2)$ doublet representation evaluated on the original mapping torus, by pullback (the singlet appearing in the decomposition does not couple to the $SU(2)$ connection, and so plays no role here). So we learn that the $SU(2)$ doublet has a $\mathbb{Z}/2$-valued global anomaly, thus providing us with a suitable generator of the anomaly group $H^6_{I\mathbb{Z}}(MT(\mathrm{Spin} \times SU(2)))$, and moreover that the pullback map from the local $SU(3)$ anomaly must be (6.5).

One can then deduce the more general condition for $SU(2)$ anomaly cancellation, by studying the equivalent condition for anomaly cancellation in the $SU(3)$ theory,

$$\mathcal{A}_{SU(3)} = 0 \bmod 2 \,. \tag{6.6}$$

Now, the irreducible representation $(n, 0)$ of $SU(3)$ decomposes into irreducible representations of $SU(2)$ as

$$(n, 0) \to \mathbf{1} \oplus \mathbf{2} \oplus \ldots \oplus \mathbf{n} \oplus (\mathbf{n+1}) \,. \tag{6.7}$$

When we decompose an $SU(3)$ representation $R := (n-1, 0) \oplus (0, n-2)$, the $SU(2)$ irreducible representations $\mathbf{1}, \mathbf{2}, \ldots, \mathbf{n-1}$ (here labelled by their dimensions) thus appear in pairs, and so cannot contribute to a mod 2 global anomaly once the anomaly is pulled back to $SU(2)$; only the irreducible representation $\mathbf{n}$ remains unpaired and so possibly contributes to the $SU(2)$ anomaly. Using the anomaly interplay map (6.5), a left-handed fermion in the representation

**n** of $SU(2)$ therefore contributes to the global $SU(2)$ anomaly if and only if the $SU(3)$ anomaly coefficient for the representation $R$ is odd. Using (6.4), this anomaly coefficient is

$$\mathcal{A}(R) := \mathcal{A}(n-1,0) + \mathcal{A}(0,n-2) = \frac{1}{12}n^2(n^2-1), \tag{6.8}$$

which is odd if and only if the dimension of the $SU(2)$ representation $n \equiv 2$ mod 4. Equivalently, in terms of isospin $j = (n-1)/2$, a left-handed fermion in the isospin $j$ representation of SU(2) contributes nontrivially to the global anomaly if and only if $j \in 2\mathbb{Z}_{\geq 0} + 1/2$. We thus reproduce the general condition for $SU(2)$ anomaly cancellation by evaluating a single local anomaly in $SU(3)$ using the APS index theorem (for the triplet representation), supplemented by basic representation theory arguments.

# 7 4d discrete gauge anomalies

For our next examples we turn to anomalies in 4d theories with discrete internal symmetries. This story goes back to pioneering work by Ibañez and Ross [20,21] on the (necessarily global) anomalies that can afflict a discrete $\mathbb{Z}/k$ gauge theory, which they derived by embedding $\mathbb{Z}/k$ in $U(1)$. Much more recently, these discrete gauge anomalies were rigorously derived by computing $\eta$-invariants by Hsieh [22], providing an intrinsic description of the global anomaly that does not rely on any microscopic completion in a $U(1)$ gauge symmetry.

In the context of the present paper, the version of this story as told by Ibañez and Ross can be understood as an instance of anomaly interplay. In this Section we recast the relation between the local $U(1)$ anomalies and the global $\mathbb{Z}/k$ anomalies from the bordism perspective, by using the calculations of Hsieh [22] to write down the precise pullback map between the anomaly theories. This pullback map from $U(1)$ anomalies to $\mathbb{Z}/k$ anomalies turn out to be surjective, meaning that one can derive necessary and sufficient conditions for cancelling the discrete gauge anomalies starting from the local $U(1)$ anomalies. In this sense, we suggest that an anomaly interplay reminiscent of Ibañez and Ross's original method does in fact give both necessary and sufficient conditions for a discrete gauge symmetry to be anomaly-free.[26]

Assuming a fermionic theory without time-reversal symmetry, an internal discrete symmetry group $K \equiv \ker \rho_n = \mathbb{Z}/k$ accommodates two possible symmetry types $(H_n, \rho_n)$ when $k = 2m$ is even. These are

$$H_n = \mathrm{Spin}(n) \times \mathbb{Z}/2m, \tag{7.1}$$

or

$$H_n = \frac{\mathrm{Spin}(n) \times \mathbb{Z}/2m}{\mathbb{Z}/2}, \tag{7.2}$$

where the $\mathbb{Z}/2$ quotient identifies the central element $(-1)^F \in \mathrm{Spin}(n)$ with the order-2 central element in $\mathbb{Z}/2m$ (that is, the element $m \cdot q$ where $q$ is a generator for $\mathbb{Z}/2m$). We will consider global anomalies for both these symmetry types, as was analyzed in [22]. In the former case we will study the interplay with local anomalies in a $\mathrm{Spin} \times U(1)$ theory, while for the latter we consider the interplay with local anomalies in a theory with $\mathrm{Spin}_c$ structure.

We are content to restrict to the case where $k = 2m = 2^n$ is a power of two with $n \geq 2$, thereby streamlining the algebra somewhat, because it is for these cases that the story with the 'twisted' symmetry type (7.2) is most interesting. For an exhaustive treatment of the $\mathbb{Z}/k$ global anomalies applicable for any integer $k$, we refer the reader to §2 of [22].

---

[26]The original method of Refs. [20,21] involve more ingredients, allowing for Yukawa couplings that give masses to some of the fermions (which may be chiral with respect to the $U(1)$ symmetry) but which nonetheless respect the $\mathbb{Z}/k$ symmetry. The perspective in this paper is a little different; we consider a completely general anomaly (up to deformation) in the $U(1)$ theory and pull that back to derive a correspondingly general anomaly in a discrete subgroup, without any additional physical constraints on the fermion spectrum.

## 7.1 Spin $\times \mathbb{Z}/2^n$

Consider embedding $H = \text{Spin} \times \mathbb{Z}/2^n$ inside $H' = \text{Spin} \times U(1)$. The relevant bordism groups for $H$ are

$$\Omega_5^{\text{Spin}}(B(\mathbb{Z}/2^n)) \cong \mathbb{Z}/2^n \times \mathbb{Z}/2^{n-2}, \qquad \Omega_6^{\text{Spin}}(B(\mathbb{Z}/2^n)) = 0, \tag{7.3}$$

with the latter condition precluding any local anomalies as we expect given there are no gauge transformations connected to the identity (and there are no pure gravitational anomalies). We compute these bordism groups using the Adams sequence in Appendix A.3, noting that $\Omega_5$ was computed by other means in [22, 51] and partial results appear also in [52]. For $U(1)$ on the other hand, we have the bordism groups

$$\Omega_5^{\text{Spin}}(BU(1)) = 0, \qquad \Omega_6^{\text{Spin}}(BU(1)) \cong \mathbb{Z} \times \mathbb{Z}. \tag{7.4}$$

(see *e.g.* Section 3.3 of [23] for $\Omega_5$, and *e.g.* §3.1.5 of [53] for $\Omega_6$.) The two factors of $\mathbb{Z}$ appearing in $\text{Hom}(\Omega_6^{\text{Spin}}(BU(1)), \mathbb{Z})$ correspond to the cubic $U(1)$ anomaly $\mathcal{A}_{\text{cub}}$ and the combination $(\mathcal{A}_{\text{cub}} - \mathcal{A}_{\text{grav}})/6$ between the mixed $U(1)$-gravitational anomaly and the cubic $U(1)$ anomaly.

The reasoning is similar to that used in §5 for the $U(2)$ local anomalies, as follows. An element of $\text{Hom}(\Omega_6^{\text{Spin}}(BU(1)), \mathbb{Z})$ is the 6th degree anomaly polynomial

$$\Phi_6 = \frac{\mathcal{A}_{\text{cub}}}{3!} \left(\frac{f}{2\pi}\right)^3 - \frac{\mathcal{A}_{\text{grav}}}{24} p_1 \frac{f}{2\pi} = \frac{1}{6}\left(\mathcal{A}_{\text{cub}} - \mathcal{A}_{\text{grav}}\right)\alpha_1 + \mathcal{A}_{\text{grav}}\alpha_2, \tag{7.5}$$

where $\alpha_1 = (f/2\pi)^3 = c_1^3$ and we now choose $\alpha_2 = \frac{1}{6}(c_1^3 - p_1 c_1/4)$. Again, $\alpha_1$ and $\alpha_2$ are basis elements of $\text{Hom}(\Omega_6^{\text{Spin}}(BU(1)), \mathbb{Z})$ dual to $(\mathbb{C}P^3, c)$, the complex projective 3-space with a $U(1)$ bundle given by $c_1 = c$, and $S_1^2 \times S_1^2 \times S_1^2$, the product of three 2-spheres with charge-1 monopoles on each factor, respectively. We should therefore identify $\mathbb{Z} \times \mathbb{Z} \cong \text{Hom}(\Omega_6^{\text{Spin}}(BU(1)), \mathbb{Z})$ with

$$\mathbb{Z}\alpha_1 \oplus \mathbb{Z}\alpha_2,$$

an element of which is labelled by the following pair of independent integers,

$$(r, s) = \left(\frac{1}{6}\left(\mathcal{A}_{\text{cub}} - \mathcal{A}_{\text{grav}}\right), \mathcal{A}_{\text{grav}}\right). \tag{7.6}$$

For example, a single left-handed Weyl fermion with unit charge corresponds to a local anomaly in the deformation class $(0, 1)$.

As usual, we abuse notation and define the embedding $\pi : \mathbb{Z}/2^n \to U(1) : q \mod 2^n \mapsto \exp(2\pi i q/2^n)$. This gives rise to a map of spectra $\pi : MTH \to MTH'$ and thus a pullback diagram for the anomaly theories pertaining to the case $d = 4$,

$$\begin{array}{ccccccc}
0 & \longrightarrow & \mathbb{Z}/2^n \times \mathbb{Z}/2^{n-2} & \longrightarrow & \mathbb{Z}/2^n \times \mathbb{Z}/2^{n-2} & \longrightarrow & 0 & \longrightarrow & 0 \\
& & \big\uparrow & & \pi^* \big\uparrow & & \big\uparrow & & \\
0 & \longrightarrow & 0 & \longrightarrow & \mathbb{Z} \times \mathbb{Z} & \longrightarrow & \mathbb{Z} \times \mathbb{Z} & \longrightarrow & 0 .
\end{array} \tag{7.7}$$

As long as the map $\pi^*$ is non-zero, this diagram encodes a non-trivial anomaly interplay. We will in fact see that $\pi^*$ is surjective, allowing the complete conditions for global anomaly cancellation in the $\mathbb{Z}/2^n$ theory to be derived from local anomalies in $U(1)$, à la Ibañez and Ross.

To study this interplay, first consider a single Weyl fermion in a general representation of $U(1)$, specified by an integer charge $Q \in \mathbb{Z}$. The anomaly theory corresponds to the element

$((Q^3 - Q)/6, Q) \in \mathbb{Z} \times \mathbb{Z} \cong \text{Hom}(\Omega_6^{\text{Spin}}(BU(1)), \mathbb{Z})$. To pull this back, decompose into representations of the discrete subgroup $\mathbb{Z}/2^n$, simply $Q \mapsto q = Q \mod 2^n$, and evaluate the global anomaly for this representation by computing the exponentiated $\eta$-invariant on generators of the bordism group $\Omega_5^{\text{Spin}}(B(\mathbb{Z}/2^n)) \cong \mathbb{Z}/2^n \times \mathbb{Z}/2^{n-2}$. Using the results of [22], the evaluations of the exponentiated $\eta$-invariant for the charge $q$ representation of $\mathbb{Z}/2^n$ on two independent generators $X$ and $Y$ of $\Omega_5^{\text{Spin}}(B(\mathbb{Z}/2^n))$ are given by

$$\begin{aligned}
\exp(2\pi i \eta(q, X)) &= \exp\left(\frac{2\pi i}{k}\left(\frac{k^2 + 3k + 2}{6} q^3\right)\right) \\
&:= \exp\left(\frac{2\pi i}{k} \nu_k\left(\frac{q^3 - q}{6}, q\right)\right),
\end{aligned} \tag{7.8}$$

and

$$\begin{aligned}
\exp(2\pi i \eta(q, Y)) &= \exp\left(\frac{2\pi i}{k/4}\left(\frac{q}{2} + \frac{k^2 + 3k + 2}{12} q^3\right)\right) \\
&:= \exp\left(\frac{2\pi i}{k/4} \nu_{k/4}\left(\frac{q^3 - q}{6}, q\right)\right),
\end{aligned} \tag{7.9}$$

where it is convenient to write these and the following formulae in terms of $k = 2^n$, and where we define the functions

$$\begin{aligned}
\nu_k(r, s) &:= \frac{1}{6}(k+1)(k+2)(6r + s), \\
\nu_{k/4}(r, s) &:= \frac{1}{2}(\nu_k(r, s) + s).
\end{aligned} \tag{7.10}$$

It is easy to see that both $\nu_k((Q^3 - Q)/6, Q)$ and $\nu_{k/4}((Q^3 - Q)/6, Q)$ are integers whenever $k = 2^n > 2$ and $Q \in \mathbb{Z}$. Moreover, for two $U(1)$ charges $Q_1$ and $Q_2$ that are congruent modulo $k$, the corresponding $\nu_k$ are congruent modulo $k$, while the corresponding $\nu_{k/4}$ are congruent modulo $k/4$. Therefore mapping the element $((Q^3 - Q)/6, Q) \in \mathbb{Z} \times \mathbb{Z} \cong \text{Hom}(\Omega_6^{\text{Spin}}(BU(1)), \mathbb{Z})$ to the element $(\nu_k((Q^3 - Q)/6, Q) \mod 2^n, \nu_{k/4}((Q^3 - Q)/6, Q) \mod 2^{n-2}) \in \mathbb{Z}/2^n \times \mathbb{Z}/2^{n-2}$ is well-defined. By linearity, we deduce that the general pullback of the anomaly theory is given by

$$\begin{aligned}
\pi^* : H_{I\mathbb{Z}}^6(MT(\text{Spin} \times U(1))) &\cong \mathbb{Z}^2 \to H_{I\mathbb{Z}}^6(MT(\text{Spin} \times \mathbb{Z}/2^n)) \cong \mathbb{Z}/2^n \times \mathbb{Z}/2^{n-2} : \\
(r, s) &\mapsto (\nu_k(r, s) \mod 2^n, \nu_{k/4}(r, s) \mod 2^{n-2}),
\end{aligned} \tag{7.11}$$

which is clearly surjective. Consequently, for an arbitrary set of $\mathbb{Z}/k$ charges $q_i$, the pair of equations

$$\begin{aligned}
\nu_k(\tilde{r}, \tilde{s}) &= 0 \mod 2^n, \\
\nu_{k/4}(\tilde{r}, \tilde{s}) &= 0 \mod 2^{n-2},
\end{aligned} \tag{7.12}$$

are necessary and sufficient for global anomaly cancellation, where $\tilde{r} := \sum(q_i^3 - q_i)/6$ and $\tilde{s} := \sum q_i$, reproducing the conditions derived in [22].

## 7.2 $(\text{Spin} \times \mathbb{Z}/2^n)/(\mathbb{Z}/2)$

We now consider the non-trivial extension of the discrete symmetry $\mathbb{Z}/2^n$ by Spin, resulting in the stable structure group $H = (\text{Spin} \times \mathbb{Z}/2^n)/(\mathbb{Z}/2)$ as defined in (7.2), and which we henceforth abbreviate by Spin-$\mathbb{Z}/2^n$. The relevant bordism groups for $H$ are

$$\Omega_5^{\text{Spin-}\mathbb{Z}/2^n} \cong \mathbb{Z}/2^{n+2} \times \mathbb{Z}/2^{n-2}, \qquad \Omega_6^{\text{Spin-}\mathbb{Z}/2^n} = 0. \tag{7.13}$$

Again, the latter condition precludes any local anomalies as expected since there are no gauge transformations connected to the identity, and no purely gravitational anomalies in 4d. We compute these bordism groups using the Adams sequence in Appendix A.4, noting that $\Omega_5$ was computed by other means in [22].

On the other hand, the corresponding non-trivial extension of $U(1)$ by Spin results in the structure group $H' = \mathrm{Spin}_c$, with the relevant bordism groups [54]

$$\Omega_5^{\mathrm{Spin}_c} = 0, \qquad \Omega_6^{\mathrm{Spin}_c} \cong \mathbb{Z} \times \mathbb{Z}. \tag{7.14}$$

Similar to the case of the trivial extension above, the deformation classes of local anomalies in the 4d $\mathrm{Spin}_c$ theory are classified by a pair of independent integers, this time[27]

$$(r,s) = \left( \frac{\mathcal{A}_{\mathrm{cub}} - \mathcal{A}_{\mathrm{grav}}}{24}, \mathcal{A}_{\mathrm{grav}} \right) \in \mathrm{Hom}\left( \Omega_6^{\mathrm{Spin}_c}, \mathbb{Z} \right) \cong \mathbb{Z} \times \mathbb{Z}. \tag{7.15}$$

The different factor of 24 instead of 6 relative to Eq. (7.6) comes from the fact that fermions can only carry odd charges.

As before, the embedding $\pi : \mathbb{Z}/2^n \to U(1) : q \bmod k \mapsto \exp(2\pi i q/2^n)$ gives rise to a map of spectra $\pi : MTH \to MTH'$ and thus a pullback diagram for the anomaly theories pertaining to the case $d = 4$,

$$
\begin{array}{ccccccc}
0 \longrightarrow & \mathbb{Z}/2^{n+2} \times \mathbb{Z}/2^{n-2} & \longrightarrow & \mathbb{Z}/2^{n+2} \times \mathbb{Z}/2^{n-2} & \longrightarrow & 0 & \longrightarrow 0 \\
 & \uparrow & & \pi^* \uparrow & & \uparrow & \\
0 \longrightarrow & 0 & \longrightarrow & \mathbb{Z} \times \mathbb{Z} & \longrightarrow & \mathbb{Z} \times \mathbb{Z} \longrightarrow 0 \,.
\end{array} \tag{7.16}
$$

This diagram encodes a non-trivial anomaly interplay if the map $\pi^*$ is non-zero. Again this will be the case. Indeed $\pi^*$ will again be a surjection. This means that the most general conditions for global anomaly cancellation in the Spin-$\mathbb{Z}/2^n$ theory can be derived by pulling back a local anomaly in $\mathrm{Spin}_c$.

To study this interplay, first consider a single Weyl fermion in a general representation of $U(1)$, specified by a charge $Q \in 2\mathbb{Z}+1$ which now must be an odd integer. The anomaly theory corresponds to the element $((Q^3 - Q)/24, Q) \in \mathbb{Z} \times \mathbb{Z} \cong \mathrm{Hom}(\Omega_6^{\mathrm{Spin}_c}, \mathbb{Z})$. To pull this back, decompose into representations of the discrete subgroup $\mathbb{Z}/2^n$, here simply $Q \mapsto q = Q \bmod 2^n$, and evaluate the global anomaly for this representation by computing the exponentiated $\eta$-invariant on generators of the bordism group $\Omega_5^{\mathrm{Spin}\text{-}\mathbb{Z}/2^n} \cong \mathbb{Z}/2^{n+2} \times \mathbb{Z}/2^{n-2}$. Using the results of [22], the evaluations of the exponentiated $\eta$-invariant for the charge $q$ representation of $\mathbb{Z}/2^n$ on two independent generators $\tilde{X}$ and $\tilde{Y}$ of $\Omega_5^{\mathrm{Spin}\text{-}\mathbb{Z}/2^n}$ are given by

$$
\begin{aligned}
\exp(2\pi i \eta(q,\tilde{X})) &= \exp\left( \frac{2\pi i}{4k} \frac{1}{12} \left( (k^2 + k + 2)q^3 - (k+6)q \right) \right) \\
&:= \exp\left( \frac{2\pi i}{4k} \mu_{4k} \left( \frac{q^3 - q}{24}, q \right) \right),
\end{aligned} \tag{7.17}
$$

---

[27]The particular dual basis chosen here is given by $\alpha_1 = c_1^3/2$ and $\alpha_2 = (c_1^3 - p_1 c_1)/48$, where $c_1$ is the first Chern class of the line bundle that determines the $\mathrm{Spin}_c$ structure, such that $c_1 = w_2 \bmod 2$. Now consider the following pair of $\mathrm{Spin}_c$ 6-manifolds, $X_1$ and $X_2$. Firstly, define $X_1$ to be the disjoint union $\mathbb{C}P^3 \sqcup (\mathbb{C}P^2 \times \mathbb{C}P^1)$, equipped with a line bundle whose first Chern class is $c_1 = 2c$ on $\mathbb{C}P^3$ and $c_1 = c' - 2a$ on $\mathbb{C}P^2 \times \mathbb{C}P^1$, where $c$, $c'$, and $a$ are the canonical generators of the second integral cohomology groups of $\mathbb{C}P^3$, $\mathbb{C}P^2$, and $\mathbb{C}P^1$ respectively. Secondly, we choose $X_2$ to be $\mathbb{C}P^1 \times \mathbb{C}P^1 \times \mathbb{C}P^1$ equipped with a line bundle whose first Chern class is $c_1 = 2a_1 + 2a_2 + 2a_3$ where $a_i \in H^2(\mathbb{C}P^1; \mathbb{Z})$ is the canonical generator of the second integral cohomology group of each $\mathbb{C}P^1$ factor. The minimal pairing is then given by $\alpha_1[X_1] = 1, \alpha_2[X_1] = 0, \alpha_1[X_2] = 24, \alpha_2[X_2] = 1$.

and

$$\exp(2\pi i\eta(q,\tilde{Y})) = \exp\left(\frac{2\pi i}{k/4}\left(\frac{k+2}{16}\right)\left((1+k)q^3 - q\right)\right)$$
$$:= \exp\left(\frac{2\pi i}{k/4}\mu_{k/4}\left(\frac{q^3 - q}{24}, q\right)\right),$$

(7.18)

where again it is convenient to use $k = 2^n$, and we define

$$\mu_{4k}(r,s) := 2(k^2 + k + 2)r + \frac{1}{12}(k-2)(k+2)s,$$
$$\mu_{k/4}(r,s) := \frac{k+2}{16}\left(24r(1+k) + sk\right).$$

(7.19)

It is easy to see that both $\mu_{4k}((Q^3-Q)/24, Q)$ and $\mu_{k/4}((Q^3-Q)/24, Q)$ are integers whenever $k = 2^n > 4$ and $Q \in 2\mathbb{Z} + 1$. Moreover, for two $\text{Spin}_c$ charges $Q_1$ and $Q_2$ that are congruent modulo $k$, the corresponding $\mu_{4k}$ are congruent modulo $4k$, while the corresponding $\mu_{k/4}$ are congruent modulo $k/4$.

Therefore, a general deformation class of $\text{Spin}_c$ local anomaly $((Q^3-Q)/24, Q) \in \mathbb{Z} \times \mathbb{Z} \cong \text{Hom}(\Omega_6^{\text{Spin-}\mathbb{Z}/2^n}, \mathbb{Z})$ is pulled back to the pair of Spin-$\mathbb{Z}/2^n$ global anomalies $(\mu_{4k}((Q^3-Q)/24, Q) \bmod 2^{n+2}, \mu_{k/4}((Q^3-Q)/24, Q) \bmod 2^{n-2}) \in \mathbb{Z}/2^{n+2} \times \mathbb{Z}/2^{n-2}$. Thence, by linearity, the general pullback is given by

$$\pi^* : H_{I\mathbb{Z}}^6\left(MT\text{Spin}_c\right) \cong \mathbb{Z}^2 \to H_{I\mathbb{Z}}^6\left(MT(\text{Spin-}\mathbb{Z}/2^n)\right) \cong \mathbb{Z}/2^{n+2} \times \mathbb{Z}/2^{n-2} :$$
$$(r,s) \mapsto \left(\mu_{4k}(r,s) \bmod 2^{n+2}, \mu_{k/4}(r,s) \bmod 2^{n-2}\right),$$

(7.20)

which is again a surjection. Consequently, for a set of arbitrary Spin-$\mathbb{Z}/2^n$ charges $q_i$, the necessary and sufficient conditions for global anomaly cancellation are

$$\mu_{4k}(\tilde{r}, \tilde{s}) = 0 \bmod 2^{n+2},$$
$$\mu_{k/4}(\tilde{r}, \tilde{s}) = 0 \bmod 2^{n-2},$$

(7.21)

where $\tilde{r} := \sum(q_i^3 - q_i)/24$ and $\tilde{s} := \sum q_i$, equivalent to the conditions derived in [22].

**Example: Standard Model and the topological superconductor**

A particularly interesting special case of the Spin-$\mathbb{Z}/2^n$ anomaly interplay occurs when $n = 2$. This case is most straightforward to analyse because there is only one independent global anomaly corresponding to[28]

$$\Omega_5^{\text{Spin-}\mathbb{Z}/4} \cong \mathbb{Z}/16,$$

(7.22)

which is generated by $\mu_{16}(\tilde{r}, \tilde{s}) = 22\tilde{r} + \tilde{s}$ (with $\tilde{r}$ and $\tilde{s}$ as defined above). Any fermion coupled to a Spin-$\mathbb{Z}/4$ structure must have charge $q_i$ equal to $\pm 1 \bmod 4$ and thus $q_i^3 - q_i = 0$, contributing 0 to $\tilde{r}$ and $\pm 1$ to $\tilde{s}$. Thus the global anomaly in fact reduces to $\mu_{16}(\tilde{r}, \tilde{s}) = \tilde{s}$, which vanishes if and only if

$$\mu_{16} = n_+ - n_- = 0 \bmod 16,$$

(7.23)

where $n_\pm$ denotes the number of Weyl fermions with charge $\pm 1 \bmod 4$.

This $\mathbb{Z}/16$-valued global anomaly was studied in Ref. [23, 37] due to a connection with the Standard Model (SM) of particle physics. The key observation was that there is a linear combination of SM global $U(1)$ symmetries under which every SM fermion has a charge

---

[28]This bordism group can also be given a lower-dimensional interpretation thanks to a Smith isomorphism $\Omega_5^{\text{Spin-}\mathbb{Z}/4} \cong \Omega_4^{\text{Pin}^+}$ [37]. From the physics perspective, one can relate each 4d Weyl fermion with Spin-$\mathbb{Z}/4$ charge to a 3d Pin$^+$ Majorana fermion on a domain wall, offering an alternative way to understand this 4d global anomaly in terms of a 3d 'topological superconductor'.

equal to 1 mod 4, which if gauged could be used to define the SM using a Spin-$\mathbb{Z}/4$ structure. Specifically, the $U(1)$ charges correspond to the linear combination $X = -2Y + 5(B-L)$, where $B-L$ denotes the difference between baryon number and lepton number, and $Y$ denotes global hypercharge.

When the SM is augmented by a trio of right-handed neutrinos (which offers the simplest route to explaining the origin of neutrino oscillation data) there are $n_+ = 16$ Weyl fermions within each generation, meaning that condition (7.23) is satisfied and there is no global anomaly.[29] As a consequence, the SM fermions can be gapped in groups of 16 by including relevant operators that preserve the Spin-$\mathbb{Z}/4$ symmetry [56], at least in specific supersymmetric extensions for which the low-energy dynamics is explicitly calculable. This example of 'symmetric mass generation' in 4d is analogous to the Fidkowski–Kitaev mechanism for gapping 1d Majorana fermions in multiples of 8 [36].

### 7.3 Going between discrete groups

We have seen above how the pullback property of the anomaly interplay map can be used to determine the global anomaly in a theory with Spin-$\mathbb{Z}/2^n$ structure, using a local anomaly calculation in a theory with Spin$_c$ structure. A simpler exercise is to deduce how global anomalies map to themselves between theories with different discrete gauge symmetries, which can be used to put constraints on global anomalies. Similar ideas were discussed in §3 of Ref. [22].

As an example, consider the interplay between a 4d theory with symmetry structure $H = $ Spin-$\mathbb{Z}/4$, as just discussed in §7.2, and a 4d theory with symmetry structure $H' = $ Spin-$\mathbb{Z}/8$. In the latter, there is a unitary symmetry operator $U$ obeying $U^4 = (-1)^F$. We embed $H$ as a subgroup of $H'$ as $\pi : H \to H' : V \mapsto U^2$, where $V$ is the order-4 element in the $\mathbb{Z}/4$ factor of $H$ that squares to $(-1)^F$. Given this embedding, a fermion with charge 1 mod 8 with respect to $H'$ has charge 1 mod 4 with respect to $H$. Suppose that we wish to calculate the global anomaly in the Spin-$\mathbb{Z}/8$ theory, and that we already know that a charge 1 mod 4 fermion in the Spin-$\mathbb{Z}/4$ theory with structure contributes a global anomaly equal to 1 mod 16 (either by direct calculation, *e.g.* as above, or by exploiting the Smith isomorphism $\Omega_5^{\text{Spin-}\mathbb{Z}/4} \cong \Omega_4^{\text{Pin}+} \cong \mathbb{Z}/16$ and knowing that a 3d Pin$^+$ Majorana fermion has a mod 16 parity anomaly). The embedding $\pi$ therefore gives rise to the interplay diagram

$$
\begin{array}{ccccccc}
0 & \longrightarrow & \mathbb{Z}/16 & \longrightarrow & \mathbb{Z}/16 & \longrightarrow & 0 & \longrightarrow & 0 \\
& & \uparrow & & \pi^* \uparrow & & \uparrow & & \\
0 & \longrightarrow & \Omega_5^{\text{Spin-}\mathbb{Z}/8} & \longrightarrow & \Omega_5^{\text{Spin-}\mathbb{Z}/8} & \longrightarrow & 0 & \longrightarrow & 0 \, .
\end{array}
\tag{7.24}
$$

Without any direct calculation, one can immediately deduce that the most refined global anomaly in the $H'$ theory must be of order $p = 16m$ with $m$ a positive integer. If it were not so, one could have a set of fewer than 16 fermions each with charge 1 mod 8 that is anomaly-free with respect to the Spin-$\mathbb{Z}/8$ structure, but that is known to have non-vanishing mod 16 anomaly with respect to $H$. This is a contradiction because the pullback $\pi^*$ is a homomorphism and must map anomaly-free Spin-$\mathbb{Z}/8$ fermion content to anomaly-free Spin-$\mathbb{Z}/4$ content. Indeed, we have seen earlier that $\Omega_5^{\text{Spin-}\mathbb{Z}/8} \cong \mathbb{Z}/32 \times \mathbb{Z}/2$ and the most refined anomaly is of order 32 which is a multiple of 16.

---

[29]There are other ways to saturate this global anomaly without introducing three right-handed neutrinos; for example, any of the (right-handed-neutrino-less) generations of SM fermions can be made anomaly free by coupling the 15 Weyl fermions to topological degrees of freedom [55].

# 8  6d anomalies in $U(1)$ *vs.* $\mathbb{Z}/2$

For our final example we turn to six spacetime dimensions and consider a similar setup to the 2d example discussed in §4, in which a theory defined with a spin structure and a unitary $\mathbb{Z}/2$ symmetry is embedded inside one with a $U(1)$ symmetry. As before, the internal symmetries do not intertwine with the spacetime symmetry, so the symmetry types are given by $H = \mathrm{Spin} \times \mathbb{Z}/2$ and $H' = \mathrm{Spin} \times U(1)$. The analysis bears similarities with the 2d example but differs in important details.

In six dimensions, the relevant bordism groups that appear in the short exact sequence classifying anomalies for the $H$ theory are[30]

$$\Omega_7^{\mathrm{Spin}}(B\mathbb{Z}/2) \cong \mathbb{Z}/16, \quad \Omega_8^{\mathrm{Spin}}(B\mathbb{Z}/2) \cong \mathbb{Z} \times \mathbb{Z}, \tag{8.1}$$

leading to anomalies classified by the short exact sequence

$$0 \to \mathbb{Z}/16 \to \mathbb{Z}/16 \times \mathbb{Z}^2 \to \mathbb{Z}^2 \to 0. \tag{8.2}$$

As we saw in the 2d example (and in contrast with the 4d examples of §7), there are local anomalies even though the gauge group is discrete. They must be purely gravitational since $\Omega_8^{\mathrm{Spin}}(\mathrm{pt}) \cong \mathbb{Z} \times \mathbb{Z}$ implies that they appear even in the absence of any gauge bundle. The group $\mathrm{Hom}(\Omega_8^{\mathrm{Spin}}(\mathrm{pt}), \mathbb{Z}) \cong \mathbb{Z} \times \mathbb{Z}$ that detects local anomalies is generated by two integer-valued bordism invariants, which we can take to be the degree 8 anomaly polynomial

$$\Phi_8 = \frac{1}{5760}(7p_1^2 - 4p_2), \tag{8.3}$$

and the signature

$$\sigma = \frac{1}{45}(7p_2 - p_1^2),$$

where $p_1$ and $p_2$ are the first and second Pontryagin classes of the tangent bundle, respectively. However, since local anomalies due to spin-1/2 chiral fermions are captured by the anomaly polynomial $\Phi_8$ alone, the gravitational anomaly due to spin-1/2 fermions is classified by one integer only. The second integer, corresponding to the signature, is the level $k$ of a 7d gravitational Chern–Simons contribution to the effective action, $\exp(2\pi i k\sigma)$.[31]

On the other hand, the relevant bordism groups for $H'$ are

$$\Omega_7^{\mathrm{Spin}}(BU(1)) = 0, \quad \Omega_8^{\mathrm{Spin}}(BU(1)) \cong \mathbb{Z}^4, \tag{8.4}$$

where the latter can be deduced from the Atiyah–Hirzebruch spectral sequence. As above, one of the $\mathbb{Z}$ factors in $\mathrm{Hom}(\Omega_8^{\mathrm{Spin}}(BU(1)), \mathbb{Z})$ is generated by the signature, which is not realised via chiral fermion anomalies. A general local anomaly due to free chiral fermions is classified by the other three integers, which we can take to be linear combinations of (i) the $U(1)$ anomaly coefficient $\mathcal{A}_{\mathrm{gauge}}$, (ii) the mixed $U(1)$-gravitational anomaly coefficient $\mathcal{A}_{\mathrm{mix}}$, and (iii) the pure gravitational anomaly coefficient $\mathcal{A}_{\mathrm{grav}}$. For an arbitrary fermion content with $N_L$ left-handed Weyl fermions of charges $q_1, \ldots, q_{N_L}$ and $N_R$ right-handed Weyl fermions of charges

---

[30]We remark that the $\mathbb{Z}/16$-valued global anomaly that we here discuss, for 6d Weyl fermions with a unitary $\mathbb{Z}/2$ symmetry, ought to be related to a parity anomaly in 4+1 dimensions, as suggested by the existence of a Smith isomorphism $\Omega_7^{\mathrm{Spin}}(B\mathbb{Z}/2) \cong \Omega_6^{\mathrm{Pin}^-} \cong \mathbb{Z}/16$ [37].

[31]Since the signature can be written in terms of the anomaly polynomial for a gravitino, via $\sigma = \Phi_{\mathrm{gravitino}} + 21\Phi_8$, the Chern–Simons level $k$ should also be included if we were to incorporate fermions of higher spins, but it will play no role in our story.

$r_1, \ldots, r_{N_R}$, these anomaly coefficients are given by

$$
\begin{aligned}
\mathcal{A}_{\text{gauge}} &= \sum_{i=1}^{N_L} q_i^4 - \sum_{i=1}^{N_R} r_i^4, \\
\mathcal{A}_{\text{mix}} &= \sum_{i=1}^{N_L} q_i^2 - \sum_{i=1}^{N_R} r_i^2, \\
\mathcal{A}_{\text{grav}} &= N_L - N_R.
\end{aligned}
\tag{8.5}
$$

Again, we cannot take all the Weyl fermions to be left-handed because in 6 dimensions, as in 2 dimensions, conjugating a complex fermion does not flip its chirality. By a similar analysis to those in §§5 and 7, we can choose the three linear combinations to be

$$
r = \frac{\mathcal{A}_{\text{gauge}} - \mathcal{A}_{\text{mix}}}{12}, \quad s = \mathcal{A}_{\text{mix}}, \quad t = \mathcal{A}_{\text{grav}}.
\tag{8.6}
$$

We now study the anomaly interplay between $\mathbb{Z}/2$ and $U(1)$ gauge theories in 6d. As usual, the subgroup embedding $\pi : \mathbb{Z}/2 \to U(1) : 1 \bmod 2 \mapsto e^{i\pi}$ induces a map of spectra $\pi : MTH \to MTH'$ as well as a pullback diagram for the anomaly theories,

$$
\begin{array}{ccccccccc}
0 & \longrightarrow & \mathbb{Z}/16 & \longrightarrow & \mathbb{Z}/16 \times \mathbb{Z}^2 & \longrightarrow & \mathbb{Z}^2 & \longrightarrow & 0 \\
& & \uparrow & & \pi^* \uparrow & & \uparrow & & \\
0 & \longrightarrow & 0 & \longrightarrow & \mathbb{Z}^4 & \longrightarrow & \mathbb{Z}^4 & \longrightarrow & 0,
\end{array}
\tag{8.7}
$$

which encodes an anomaly interplay between a $U(1)$ local anomaly and a global anomaly in the $\mathbb{Z}/2$ theory through the pullback $\pi^*$. Since the difference between the pair of symmetry types $H$ and $H'$ does not involve spacetime symmetry, the pure gravitational anomalies maps to themselves under $\pi^*$, playing no role in the interplay.

To see how the pullback acts on the remaining two factors of $\mathbb{Z}$ in $H^8_{I\mathbb{Z}}(MT(\text{Spin} \times U(1)))$, we consider a generic fermion spectrum coupled to a 6d $U(1)$ gauge field, the charges of which we parametrize as in §4. The gravitational anomaly and the mixed $U(1)$-gravitational anomaly cancellation conditions, $\mathcal{A}_{\text{grav}} = \mathcal{A}_{\text{mix}} = 0$, are given by Eqns. (4.6) and (4.7), respectively. In addition to these two conditions (which are of the same form as in the 2d case), we also require that the quartic $U(1)$ anomaly vanishes, $\mathcal{A}_{\text{gauge}} = 0$, which implies

$$
16 \left( \sum_{i=1}^{N_L^e} k_i^4 - \sum_{i=1}^{N_R^e} k_i'^4 \right) + 16 \sum_{i=1}^{N_L^o} \frac{1}{2} l_i(l_i+1)(2l_i^2 + 2l_i + 1)
$$
$$
- 16 \sum_{i=1}^{N_R^o} \frac{1}{2} l_i'(l_i'+1)(2l_i'^2 + 2l_i' + 1) + N_L^o - N_R^o = 0,
\tag{8.8}
$$

and hence that the index for the oddly-charged Weyl fermions must be a multiple of 16, *viz.*

$$
N_L^o - N_R^o \in 16\mathbb{Z}.
\tag{8.9}
$$

The $\mathbb{Z}/2$ subgroup of this $U(1)$ gauge group only acts non-trivially on fermions with odd charge. Only these odd-charged fermions contribute to both the $\mathbb{Z}/16$ global anomaly encoded in (8.2) and the gravitational anomaly, while the even-charged fermions can be used to soak up the gravitational anomaly. Since $\pi^*$ is a homomorphism it maps zero to zero, and thus maps any anomaly-free spectrum to another. So if a single left-handed, $\mathbb{Z}/2$-charged Weyl fermion contributes an anomaly equal to $\nu$ mod 16, then our considerations in the previous paragraph

imply that $16\nu = 0$ mod 16. Thus, it is permissible on the ground of anomaly interplay that a single left-handed odd Weyl fermion contributes the most refined anomaly of $\nu = 1$ mod 16.

This is indeed the case, as can be seen by explicit computation of the exponentiated $\eta$-invariant on a generator of the bordism group $\Omega_7^{\text{Spin}}(B\mathbb{Z}/2)$. To wit, for a left-handed odd-charged Weyl fermion together with a right-handed even-charged Weyl fermion (thus cancelling any local anomaly, making the exponentiated $\eta$-invariant a bordism invariant), one finds $\exp(2\pi i\eta_X) = \exp(2\pi i/16)$ where $X$ is $\mathbb{R}P^7$ equipped with a $\mathbb{Z}/2$ gauge bundle whose first Stiefel–Whitney class equals the generator of $H^1(\mathbb{R}P^7; \mathbb{Z}/2)$, which is a generator of $\Omega_7^{\text{Spin}}(B\mathbb{Z}/2)$ (c.f. §3.6 of Ref. [37]).[32]

To conclude our analysis of 6d anomalies, the anomaly pullback map $\pi^*$ is here given by

$$\pi^* : H^8_{I\mathbb{Z}}(MT(\text{Spin} \times U(1))) \cong \mathbb{Z}^4 \to H^8_{I\mathbb{Z}}(MT(\text{Spin} \times \mathbb{Z}/2)) \cong \mathbb{Z}/16 \times \mathbb{Z}^2 :$$
$$(r, s, t, k) \mapsto ((12r + s) \bmod 16, t, k) . \tag{8.10}$$

Note that, unlike in the 2d case, the pullback map $\pi^*$ is surjective. This is permissible because in 6 dimensions there is no chiral basis for the gamma matrices where all elements are real. Hence, a Weyl fermion cannot be divided further into two real chiral fermions as in 2 dimensions.

# Acknowledgments

We thank Pietro Benetti Genolini, Philip Boyle Smith, Ben Gripaios, Avner Karasik, David Tong, and Carl Turner for many useful discussions about this project. We especially thank Philip Boyle Smith for providing helpful comments on the manuscript. NL is supported by David Tong's Simons Investigator Award. We are supported by the STFC consolidated grant ST/P000681/1.

# A   Some bordism calculations

In this Appendix, we sketch how bordism groups are calculated using the Adams spectral sequence [57, 58], and then present the calculations of some bordism groups mentioned in the main text. For a more detailed introduction to practical calculations using the Adams spectral sequence, we recommend Ref. [14]. All cohomology rings have coefficients in $\mathbb{Z}/2$ unless otherwise stated.

## A.1   Using the Adams spectral sequence

We will use the Adams spectral sequence to obtain the 2-completion $\left(\Omega_{t-s}^H\right)_2^\wedge$ of a given bordism group $\Omega_{t-s}^H$ that we wish to calculate. Recall that the 2-completion of the group of integers $\mathbb{Z}$ is the 2-adic group $\mathbb{Z}_2$, the 2-completion of the cyclic group $\mathbb{Z}/2^n$ is $\mathbb{Z}/2^n$, while the 2-completion of $\mathbb{Z}/m$ when $m$ is odd is the trivial group. Therefore, if there is no odd torsion involved, the whole bordism group can be obtained from its 2-completion.

The second page of the Adams sequence then has entries

$$E_2^{s,t} = \text{Ext}_{\mathcal{A}_2}^{s,t}(H^\bullet(MTH), \mathbb{Z}/2) \Rightarrow \left(\Omega_{t-s}^H\right)_2^\wedge , \tag{A.1}$$

---

[32]In principle, it must be possible to *derive* this expression for $\exp(2\pi i\eta_X)$ (and thence the $\mathbb{Z}/16$-valued global anomaly) by embedding the $\mathbb{Z}/2$ connection inside a $U(1)$ connection, which can be extended to an 8-manifold bounding $\mathbb{R}P^7$ because $\Omega_7^{\text{Spin}}(BU(1)) = 0$. Having performed a similar analysis in the analogous 3d setting in §4, we are content to omit an explicit calculation here.

where $\mathcal{A}_2$ is the Steenrod algebra and $MTH$ is the Madsen–Tillmann spectrum of a stable tangential structure group $H$. If the spectrum $MTH$ can moreover be written as $M\mathrm{Spin} \wedge X_H$, the Anderson–Brown–Peterson theorem means that (A.1) simplifies to

$$E_2^{s,t} = \mathrm{Ext}_{\mathcal{A}(1)}^{s,t}(H^{\bullet}(X_H), \mathbb{Z}/2) \Rightarrow \left(\Omega_{t-s}^{H}\right)_2^{\wedge}, \tag{A.2}$$

for $t - s < 8$, where $\mathcal{A}(1)$ is the subalgebra of $\mathcal{A}_2$ generated by the Steenrod squares $\mathrm{Sq}^1$ and $\mathrm{Sq}^2$. Fortunately, this simplification occurs in all of the examples we consider in this Appendix. For example, when $H = \mathrm{Spin} \times G$ is a product of the stable spin group and an internal symmetry group $G$, we have $MTH = M\mathrm{Spin} \wedge BG_+$, where $+$ denotes a disjoint basepoint. In this case $\Omega_{t-s}^{H}$ is the $(t-s)^{\text{th}}$ spin bordism group of $BG$, often denoted by $\Omega_{t-s}^{\mathrm{Spin}}(BG)$.

The next step is to compute the $\mathcal{A}(1)$-module structure of $H^{\bullet}(X_H)$, and plot the corresponding graded extension in an Adams chart. The Adams chart for $H^{\bullet}(X_H)$ is a visual representation of $\mathrm{Ext}_{\mathcal{A}(1)}^{s,t}(H^{\bullet}(X_H), \mathbb{Z}/2)$ on the $(t-s,s)$-plane, in which a dot represents a generator of $\mathbb{Z}/2$. There can be some non-trivial relations between generators of $E_2^{s,t}$. The ones that we will encounter in the sequel are a) multiplication[33] by an element $h_0 \in \mathrm{Ext}_{\mathcal{A}(1)}^{1,1}(\mathbb{Z}/2, \mathbb{Z}/2)$, which is represented by a vertical line segment, and b) multiplication by an element $h_1 \in \mathrm{Ext}_{\mathcal{A}(1)}^{1,2}(\mathbb{Z}/2, \mathbb{Z}/2)$, which is represented by a line segment of slope 1. Being a spectral sequence, each page $E_r$ is a bi-graded complex of abelian groups, equipped with group homomorphisms or 'differentials'

$$d_r : E_r^{s,t} \to E_r^{s+r,t+r-1}, \quad d_r \circ d_r = 0.$$

Pictorially, in the Adams chart each of these differentials is represented by an arrow that goes back one column to the left, and up by $r$ rows. These differentials commute with multiplication by $h_0$. Turning to the next page of the Adams sequence, an element $E_{r+1}^{s,t}$ is obtained by taking the homology of the complex $E_r$ at $E_r^{s,t}$. As one continues to turn the pages, the elements of the Adams charts stabilise to what we will denote by $E_{\infty}^{s,t}$.

The existence of the Adams spectral sequence means that there is a filtration of $\left(\Omega_t^{H}\right)_2^{\wedge}$ given by

$$\left(\Omega_t^{H}\right)_2^{\wedge} = F_{\infty}^{0,t} \supseteq F_{\infty}^{1,t+1} \supseteq F_{\infty}^{2,t+2} \supseteq \dots \tag{A.3}$$

The last page $E_{\infty}$ of the Adams chart encodes this filtration as follows. The quotient $F_{\infty}^{0,t}/F_{\infty}^{s,t+s}$ of $\left(\Omega_t^{H}\right)_2^{\wedge}$ by its filtered sets can be found inductively by solving the group extension problem

$$0 \longrightarrow E_{\infty}^{s,t+s} \longrightarrow F_{\infty}^{0,t}/F_{\infty}^{s+1,t+s+1} \longrightarrow F_{\infty}^{0,t}/F_{\infty}^{s,t+s} \longrightarrow 0. \tag{A.4}$$

This information can be formally assembled to obtain $\left(\Omega_t^{H}\right)_2^{\wedge}$, namely by taking the inverse limit

$$\left(\Omega_t^{H}\right)_2^{\wedge} \cong \varprojlim_s F_{\infty}^{0,t}/F_{\infty}^{s,t+s}. \tag{A.5}$$

Finally, it is worth remarking that the extension problem just described can sometimes be non-trivial in a controlled way, determined by the module structure on the $E_2$ page of the Adams chart. For example, a multiplication by $h_0$ on the $E_2$ page records multiplication by 2 between $F_{\infty}^{s,t}$ and $F_{\infty}^{s+1,t+1}$. Thus, an infinite $h_0$-tower in the $t$ column implies $F_{\infty}^{0,t}/F_{\infty}^{s,t+s} \cong \mathbb{Z}/2^s$ for all $s$, which gives $F_{\infty}^{0,t} \cong \mathbb{Z}_2$. Similarly, if there is a truncated $h_0$-tower of length $m$, we get $F_{\infty}^{0,t}/F_{\infty}^{s,t+s} \cong \mathbb{Z}/2^s$ for $s \leq m$ and remains $\mathbb{Z}/2^m$ when $s > m$; it can be easily seen that $F_{\infty}^{0,t} = \mathbb{Z}/2^m$ in this case. In most cases considered here, these are all the non-trivial extensions

---

[33]By 'multiplication' we here refer to the Yoneda product $\mathrm{Ext}_{\mathcal{A}(1)}^{s,t}(M, \mathbb{Z}/2) \otimes_{\mathbb{Z}/2} \mathrm{Ext}_{\mathcal{A}(1)}^{s',t'}(\mathbb{Z}/2, \mathbb{Z}/2) \to \mathrm{Ext}_{\mathcal{A}(1)}^{s+s',t+t'}(M, \mathbb{Z}/2)$ for an $\mathcal{A}(1)$-module $M$.

that appear. So when the bordism groups can be fully calculated at the 2-completion we can read off the results from the Adams chart directly, whence an infinite $h_0$-tower gives the free Abelian group $\mathbb{Z}$, and a truncated $h_0$-tower of length $m$ gives the torsion group $\mathbb{Z}/2^m$. There can also be non-trivial extensions that do not arise from the structure of the $E_2$ page. These are called exotic extensions.

## A.2 Calculation of $\Omega_6^{\mathrm{Spin}\text{-}U(2)}$

It was calculated in Ref. [16] that the associated Madsen–Tillmann spectrum for the symmetry type $H = \mathrm{Spin}\text{-}U(2)$ is $MTH = M\mathrm{Spin} \wedge \Sigma^{-5} MSO(3) \wedge MU(1)$. The second page of the Adams spectral sequence is then given by

$$E_2^{s,t} = \mathrm{Ext}_{\mathcal{A}(1)}^{s,t}\left(\Sigma^{-5} H^\bullet(MSO(3) \wedge MSO(2)), \mathbb{Z}/2\right). \tag{A.6}$$

The cohomology ring here is

$$H^\bullet(MSO(3) \wedge MSO(2)) \cong \mathbb{Z}/2[w_2', w_3', w_2''] \, UV,$$

where $w_i' \in H^i(BSO(3))$ are the universal Stiefel–Whitney classes for $SO(3)$ while $w_2'' \in H^2(BSO(2))$ is the second universal Stiefel–Whitney class for $SO(2)$. Here $U \in H^3(MSO(3))$ and $V \in H^2(MSO(2))$ are the Thom classes for $MSO(3)$ and $MSO(2)$, respectively.

The $\mathcal{A}(1)$-module structure of this ring, up to degree 10, was also calculated in Ref. [16]. This module structure is represented by Fig. 2. This was computed by applying the 'Wu formula'

$$\mathrm{Sq}^i w_j = \sum_{k=0}^i \binom{j-i+k-1}{k} w_{i-k} \, w_{j+k}, \tag{A.7}$$

for the action of the Steenrod squares on Stiefel–Whitney classes, together with the relations $\mathrm{Sq}^2 U = w_2' U$, $\mathrm{Sq}^2 V = w_2''$, and $\mathrm{Sq}^1 U = \mathrm{Sq}^1 V = 0$. Since there are only 5 generators in $H^{11}(MSO(3) \wedge MSO(2))$ and 3 generators in $H^{12}(MSO(3) \wedge MSO(2))$, the diagram in Fig. 2 in fact represents the module $H^\bullet(MSO(3) \wedge MSO(2))$ up to degree 12, so it can be used to calculate the Adams chart (and thence the bordism groups) up to degree $t - s = 7$, which is reproduced in Fig. 3. The $6^{\text{th}}$ bordism group can then be read off directly from the chart, giving

$$\Omega_6^{\mathrm{Spin}\text{-}U(2)} \cong \mathbb{Z} \times \mathbb{Z} \times \mathbb{Z}. \tag{A.8}$$

## A.3 Calculation of $\Omega_d^{\mathrm{Spin}}(B\mathbb{Z}/2^m)$ for $m > 1$

To compute the bordism groups for the symmetry type $\mathrm{Spin} \times B\mathbb{Z}/2^m$, our first task is to work out the $\mathcal{A}(1)$-module structure of $H^\bullet(B\mathbb{Z}/2^m)$. It is known that

$$H^\bullet(B\mathbb{Z}/2^m) \cong H^\bullet(K(\mathbb{Z}/2^m, 1)) \cong \mathbb{Z}/2[a, b]/(a^2), \tag{A.9}$$

where the generator $a$ is in degree 1, and the generator $b$ is in degree 2 [59] (see also Theorem 6.19 of [60]). The generator $b$ can be defined as follows. The short exact sequence

$$0 \longrightarrow \mathbb{Z}/2 \longrightarrow \mathbb{Z}/2^{m+1} \longrightarrow \mathbb{Z}/2^m \longrightarrow 0, \tag{A.10}$$

of coefficient groups induces the long exact sequence in cohomology

$$\dots \to H^i(X; \mathbb{Z}/2) \to H^i(X; \mathbb{Z}/2^{m+1}) \to H^i(X; \mathbb{Z}/2^m) \xrightarrow{\beta_m} H^{i+1}(X; \mathbb{Z}/2) \to \dots, \tag{A.11}$$

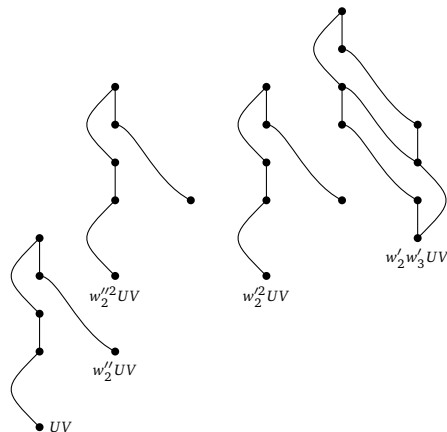

**Figure 2:** The $\mathcal{A}_1$-module structure for $\mathbb{Z}/2[w_2', w_3', w_2'']\{UV\}$, up to degree 12.

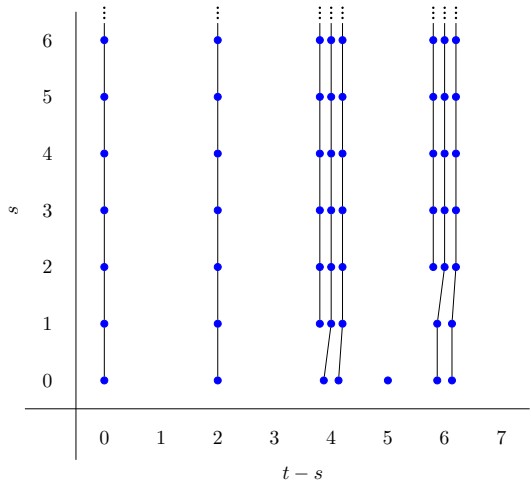

**Figure 3:** The Adams chart for $\Sigma^{-5}H^{\bullet}(MSO(3) \wedge MSO(2))$ with $t-s \le 7$.

for any topological space $X$ (here we temporarily restore the explicit $\mathbb{Z}/2$ coefficients for emphasis). The connecting homomorphism $\beta_m$ is the $m^{\text{th}}$ power Bockstein homomorphism. One can then define $b = \beta_m(a')$, where $a' \in H^1(B\mathbb{Z}/2^m; \mathbb{Z}/2^m)$ is a canonical choice of generator.

The $\mathcal{A}(1)$-module structure of $H^{\bullet}(B\mathbb{Z}/2^m)$ is then given in Fig. 6, where the dashed lines represent the $m^{\text{th}}$ power Bockstein $\beta_m$. Hence, as an $\mathcal{A}(1)$-module, we can write $H^{\bullet}(B\mathbb{Z}/2^m)$ as

$$H^{\bullet}(B\mathbb{Z}/2^m) = \mathbb{Z}/2 \oplus \Sigma\mathbb{Z}/2 \oplus \Sigma^2\mathcal{M} \oplus \Sigma^3\mathcal{M} \oplus \Sigma^6\mathcal{M} \oplus \Sigma^7\mathcal{M}\dots, \qquad (A.12)$$

where the $\mathcal{A}(1)$-module $\mathcal{M}$ together with its corresponding Adams chart, and the Adams chart for the $\mathcal{A}(1)$-module $\mathbb{Z}/2$, are shown in Figs. 4 and 5, respectively. We can then combine these to construct the Adams chart for $H^{\bullet}(B\mathbb{Z}/2^m)$, shown in Fig. 7 for $m = 3$ as an example. By the May–Milgram Theorem [61], the only non-trivial differentials are those denoted by $d_m$, which are induced by the Bockstein homomorphism on the classes with even $t-s$. From the Adams chart it is clear that $\Omega_6^{\text{Spin}}(B\mathbb{Z}/2^m) = 0$. The resulting spin bordism groups in degrees $d \le 6$ are given in Table 1. Note that there are non-trivial extensions in degree 5, whose corresponding bordism group has been calculated by other means in Refs. [22, 51].

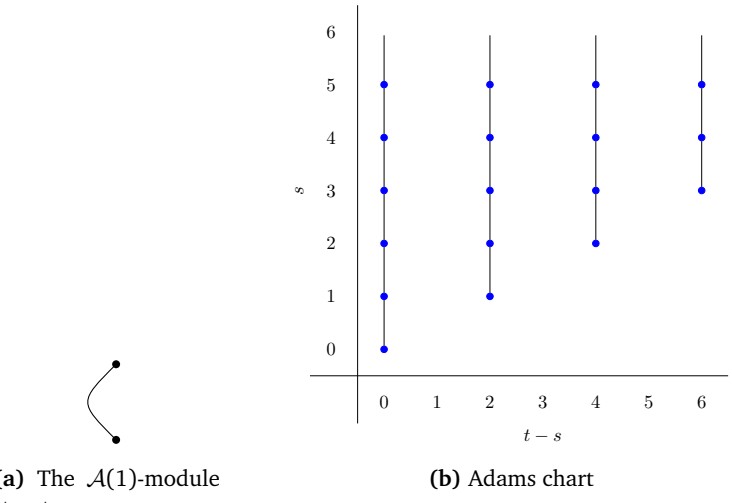

**(a)** The $\mathcal{A}(1)$-module structure.

**(b)** Adams chart

**Figure 4:** The $\mathcal{A}(1)$-module $\mathcal{M}$.

**Figure 5:** The Adams chart for $\mathbb{Z}/2$.

**Table 1:** The spin bordism groups for $B\mathbb{Z}/2^m$ in degrees $d \leq 6$ for $m > 1$.

| $d$ | 0 | 1 | 2 | 3 | 4 | 5 | 6 |
|---|---|---|---|---|---|---|---|
| $\Omega_d^{\mathrm{Spin}}(B\mathbb{Z}/2^m)$ | $\mathbb{Z}$ | $\mathbb{Z}/2 \times \mathbb{Z}/2^m$ | $\mathbb{Z}/2 \times \mathbb{Z}/2$ | $\mathbb{Z}/2 \times \mathbb{Z}/2^{m+1}$ | $\mathbb{Z}$ | $\mathbb{Z}/2^m \times \mathbb{Z}/2^{m-2}$ | 0 |

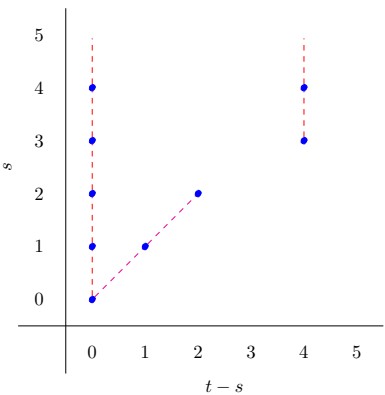

**Figure 6:** The ring $H^\bullet(B\mathbb{Z}/2^m)$ as an $\mathcal{A}(1)$-module.

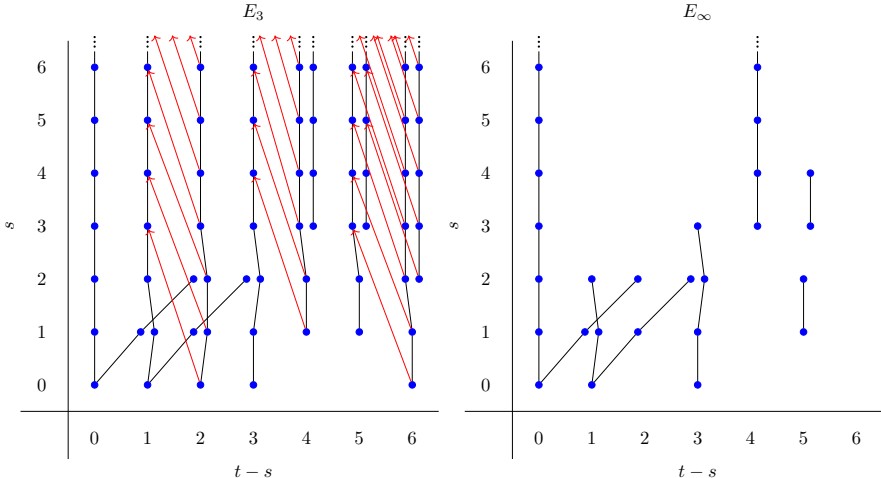

**Figure 7:** The Adams chart for $H^\bullet(B\mathbb{Z}/2^3)$.

## A.4 Calculation of $\Omega_d^{\text{Spin-}\mathbb{Z}/2^{m+1}}$ for $m > 1$

Finally, when the stable symmetry type is $H = \text{Spin-}\mathbb{Z}/2^{m+1}$, the Madsen–Tillman spectrum $MTH$ is [13]

$$MTH = M\text{Spin} \wedge (B(\mathbb{Z}/2^m))^{2\xi}, \tag{A.13}$$

where $2\xi$ is twice the sign representation and $(B(\mathbb{Z}/2^m))^{2\xi}$ is the Thom space of the bundle $2\xi$. Now, we have to work out the $\mathcal{A}(1)$-module structure of $H^\bullet(B(\mathbb{Z}/2^m))^{2\xi}$, which turns out to be the same as that of $H^\bullet(B\mathbb{Z}/2^m)$ but with the two bottom cells removed, as shown in Fig. 8. Hence, as an $\mathcal{A}(1)$-module we can write

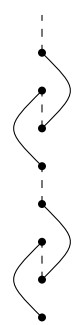

**Figure 8:** The ring $H^\bullet((B\mathbb{Z}/2^m)^{2\xi})$ as an $\mathcal{A}(1)$-module.

$$H^\bullet(B(\mathbb{Z}/2^m))^{2\xi} = \mathcal{M} \oplus \Sigma\mathcal{M} \oplus \Sigma^4\mathcal{M} \oplus \Sigma^5\mathcal{M} \oplus \Sigma^8\mathcal{M} \oplus \Sigma^9\mathcal{M}\ldots, \tag{A.14}$$

where the $\mathcal{A}(1)$-module $\mathcal{M}$ is defined as before (see Figure 4a). Using the Adams charts for $\mathcal{M}$ given in Fig. 4b, we can see that the second page of the Adams chart for $H^\bullet(B(\mathbb{Z}/2^m))^{2\xi}$ must be given by Fig. 9.

As in the previous case of $H^\bullet(B\mathbb{Z}/2^m)$, the only non-trivial differentials are $d_m$ on the $m^{\text{th}}$ page, which are induced from the $m^{\text{th}}$ power Bockstein homomorphism [13,61]. So the non-trivial differentials act only on the even $t-s$ columns. As an example, these are shown for $m = 2$ in Fig. 10. This can be easily generalised to $m \geq 2$, with results shown in Table 2. The $5^{\text{th}}$ bordism group was calculated by a different method in Ref. [22].

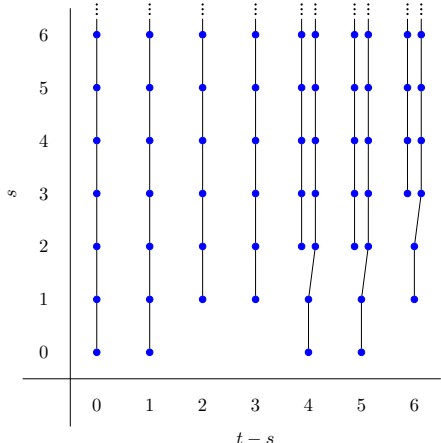

**Figure 9:** The Adams chart for $H^\bullet(B(\mathbb{Z}/2^m))^{2\xi}$.

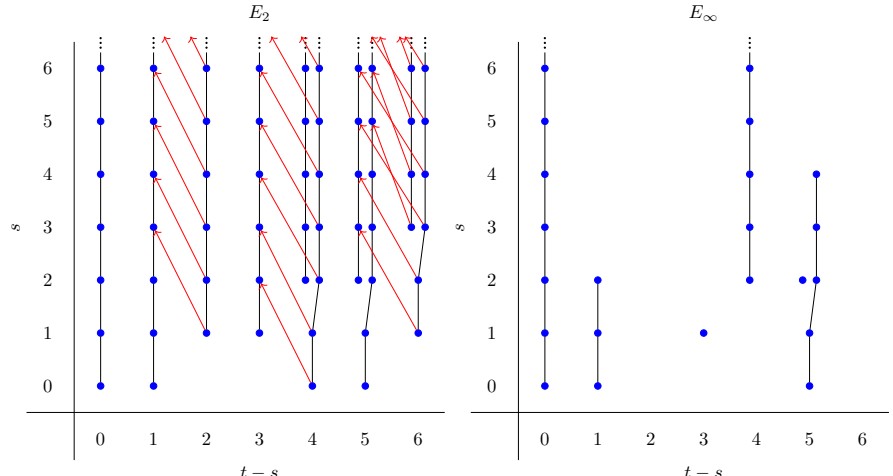

**Figure 10:** The $E_2$ and $E_\infty$ pages of the Adams chart for $H^\bullet(B(\mathbb{Z}/2^2))^{2\xi}$.

**Table 2:** The bordism groups with Spin-$\mathbb{Z}/2^{m+1}$ structure in degrees $d \le 6$ for $m > 1$.

| $d$ | 0 | 1 | 2 | 3 | 4 | 5 | 6 |
|---|---|---|---|---|---|---|---|
| $\Omega_d^{\text{Spin-}\mathbb{Z}/2^{m+1}}$ | $\mathbb{Z}$ | $\mathbb{Z}/2^{m+1}$ | 0 | $\mathbb{Z}/2^{m-1}$ | $\mathbb{Z}$ | $\mathbb{Z}/2^{m-1} \times \mathbb{Z}/2^{m+3}$ | 0 |

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
