# Peer review of "The algebra of anomaly interplay"

_SciPost Physics, doi:SciPost Phys. 10, 074 (2021)_

## Round 2 · Referee Report · Anonymous (Referee 1) · 2021-2-4

Report

In this paper the authors study how the anomaly of a fermion system with symmetry $G$ looks when only the subgroup $H\subset G$ is considered, when $G$ can have nonzero anomaly polynomials while $H$ can only have global anomalies. This question was first studied in the 80s by Elitzur and Nair using the homotopy group of $G$, but the authors revisit this question using the modern understanding of the anomalies based on bordism groups.

The manuscript is well presented and can be recommended for publication, once the following points are taken care of.

  1. Referring to the (Anderson/Pontryagin) dual of the bordism group as the cobordism group is often done in recent physics literature but the referee considers it still as a non-standard usage which can lead to confusion. Historically, what are now called as bordism groups were called cobordism groups. (Two manifolds can be called cobordant or bordant, interchangeably.) Later, the math community shifted to call them bordism groups, most probably because cobordism groups are found to be generalized homology groups (rather than generalized cohomology groups) associated to Thom spectra. One could have then repurposed the name "cobordism groups" to the generalized cohomology theories associated to Thom spectra, but apparently it wasn't done very often.

  2. The authors use the symbol $H_n$ for two closely related concepts concerning symmetry types, which the referee found too confusing. Namely, $H_n$ is a sequence of groups specifying the symmetry type, whose limit is $H$. Then, the authors use $H_1$ and $H_2$ to refer to limits of two such sequences $(H_1)_n$ and $(H_2)_n$ respectively. This makes it unclear what is meant when the author simply writes $H_i$. The referee would suggest the authors to use $H$ and $H'$ instead for the second purpose.

  3. in p.5, the authors refer to "the Anderson dual of the sphere spectrum $I\mathbb{Z}$." Those who are unfamiliar with these matters won't be able to tell whether $I\mathbb{Z}$ is the sphere spectrum or the Anderson dual of that. It would be better to write the Anderson dual $I\mathbb{Z}$ of the sphere spectrum.

  4. Later in p.5, the authors say gravitational Chern-Simons terms in even dimensions are obvious counterexamples, but why? In the 2d/3d case, a chiral 2d fermion can live on the boundary of the gravitational Chern-Simons term, and similarly in higher dimensions, at least for a particular choice of gravitational Chern-Simons terms corresponding to the $\hat A$ genus.

  5. In p.7 footnote 7, the authors say that the appearance of line bundles is what relates the fermion anomaly to K theory, but this is doubtful. Many of the discussions in [24] or [3] applies to any combination of the boundary theory and the anomaly theory, resulting in the fact that the partition function of the boundary theory is a section of a line bundle which is specified by the anomaly theory. This is independent of whether the theory in question is a free fermion theory or not. Yes the K-theory is defined in terms of vector bundles but it requires not just line bundles but vector bundles with general dimensions.

  6. This is simply a comment. In p.12, the authors say that Elitzur and Nair preceded the modern understanding of global anomalies via bordisms. But Elitzur-Nair is from 1984 while Witten's "Global Gravitational Anomalies" is from 1985, and the latter essentially contains the statement that the global anomalies are captured by bordism invariants whose prime examples are eta. So it is not perfectly correct to say that the understanding of (global) anomalies via bordisms is modern. It is just that its popularization is modern.

  7. In Sec 4 and later, the authors provide many explicit examples. There is one general problem in the presentation, when the authors describe the generators of the free part of the extension sequence, which is of the form $\mathbb{Z}^n$. This is done by the authors by finding $n$ anomaly polynomials which correctly integrate to integers. But this argument only guarantees that the authors found a subgroup $A\subset \mathbb{Z}^n = \mathrm{Hom}(\pi_{d+2}(MTH),\mathbb{Z})$ such that $A\simeq \mathbb{Z}^n$, but $A$ can be a proper subgroup.

The referee believes this did not actually caused the problem, because of two reasons. First, the pull-back of this (possibly proper) subgroup $A$ was enough to reproduce the global anomaly in most of the cases (except in the 2d case where the $\mathbb{Z}_2$ anomaly of Majorana-Weyl fermion was not covered).

Second, the group $A$ found by the authors was actually equal to the full $\mathrm{Hom}(\pi_{d+2}(MTH),\mathbb{Z})$ in all the cases in this paper, but this is something one needs to be checked.

  1. In p.14, the authors might want to say that the splitting $\Omega(X)=\Omega(pt)+\tilde\Omega(X)$ is canonical.

  2. In p.16, the authors describe a generator of $\mathbb{Z}_8$ as a mapping torus and then find its null bordism. The referee wants to mention that the null bordism of $\mathbb{RP}^3$ where a $\mathbb{Z}_2$ connection is embedded in $U(1)$ is also standard. Namely, $\mathbb{RP}^3$ is the asymptotic infinity of the simplest type of the ALE space (originally found by Eguchi and Hanson), which is topologically a disk bundle over $S^2$. See e.g. Sec.4.1 of http://arxiv.org/abs/hep-th/9605184 . There, the integral $\int F\wedge F/(2\pi)^2=-1/4$ is also given in (4.6); note that the normalization of $F$ is different by a factor of two there.

  3. The referee did not understand the footnote 19 in p.18. Isn't it that the author considered a U(1)-charged Weyl fermion? Then there are only charged right-movers. If so, when $U(1)$ is restricted to $\mathbb{Z}_2$, it seems to simply give two copies of $\mathbb{Z}_2$-odd Majorana-Weyl fermions.

  4. In p.20, could you clarify how $\mathcal{F}$, $f$ and $F$ are related? There can be multiple conventions imaginable. Also, it was not clear why the authors use $(1/2)(f/2\pi)$ instead of $f/(2\pi)$ for the first Chern class.

  5. In p.22, a related issue is that the authors' human language explanation of the $U(1)$ flux on the hemisphere was too ambiguous. When the authors say a monopole flux on $S^2$, do they consider a spherically symmetric configuration, or a configuration concentrated on a single hemisphere? Namely, the referee could not figure out whether the authors intended to have $\int_{S^2} f=\int_{D^2} f$ or $\int_{S^2}=2\int_{D^2} f$. Also, the referee could not figure out whether the minimal Dirac flux corresponds to

    $$ \int_{S^2?\ \text{or}\ D^2} (\frac{f}{2\pi}\ \text{or}\ \frac12\frac{f}{2\pi}) = 1. $$
    Please clarify.

  6. In p.37, footnote 37, the authors say that $\sigma$ doesn't correspond to a chiral fermion. But $\sigma-21\Phi_8$ is the anomaly of a gravitino, which is a chiral fermion of spin $3/2$.

  7. In p.45, (A.13), please say that $B(\mathbb{Z}/2^m))^{2\xi}$ is the Thom space of the bundle $2\xi$.

  8. In p.51, the URL for [48] was listed twice.

  • validity: -
  • significance: -
  • originality: -
  • clarity: -
  • formatting: -
  • grammar: -

Author:  Nakarin Lohitsiri  on 2021-03-02  [id 1279]

(in reply to Report 1 on 2021-02-04)
Category:
answer to question
correction

  1. We have highlighted that our use of the word cobordism is non-standard in the mathematical literature; however, because of the prominent role of the (cohomological) dual of bordism in this paper, we wish to retain our usage of the word. To address the Referee's concerns, we added a new footnote 3, and additional sentences of clarification in Section 2 (soon after equation (2.7)), to emphasize that we use the word 'cobordism' solely to mean the (Anderson) dual of bordism, itself a generalized cohomology theory. Correspondingly, we are now careful to always use the word 'bordism' when referring to the equivalence relation on manifolds, and to the corresponding generalized homology groups.
  2. We have implemented this helpful suggestion.
  3. Amended.
  4. We have removed the sentence about gravitational Chern-Simons terms. To explain our original intentions, we had in mind not the 2d/3d case (where $p_1$ appears in the anomaly polynomial for a spin-$1/2$ fermion) but the 6d/7d case. Our original reasoning was that there are two independent gravitational Chern--Simons couplings one can write down, corresponding to $p_1^2$ and $p_2$, while the anomaly polynomial for a spin-$1/2$ fermion only generates the linear combination $7p_1^2-4p_2$. Thanks to the referee's comment 13., we now realise that an independent combination is given by the anomaly polynomial of the gravitino, and so both gravitational CS terms are indeed 'covered' by anomaly polynomials, provided one allows fermions of higher spins. We thank the reviewer for bringing this to our attention. Nonetheless, the conjecture of Freed-Hopkins, as recalled in our footnote 6 (pg 5), suggests to us that free fermion anomalies are in general a subgroup of the Anderson dual to the bordism group, and we would be very interested to know of explicit examples of invertible phases not captured by free fermions.
  5. We removed the offending footnote, and thank the reviewer for the helpful explanation.
  6. We have followed the referee's suggestion to rephrase things, and added a footnote (12) drawing attention to Witten's first paper featuring bordism invariants and $\eta$.
  7. In order to address this gap in our argument, we have now identified generators of the group $\text{Hom}(\Omega_{d+2}^H, \mathbb{Z})$ in our various examples by explicit computation, following the strategy outlined in the old footnote 22 (pg 20 of the original version). For example, in Section 5 (concerning $U(2)$ vs $SU(2)$ 4d anomalies) we have constructed three dual basis elements for $\text{Hom}(\Omega^{\text{Spin}}_6(BU(2)),\mathbb{Z})$. We checked the minimal pairing between these elements and three generators of the bordism group $\Omega^{\text{Spin}}_6(BU(2))$ (i.e. one basis element integrates to $1$ on one generator and $0$ on the others), thereby showing that our basis indeed spans the whole group. We similarly checked the $U(1)$ cases in 4d (including the Spin$_c$ version) and 6d. The text in these Sections has been substantially expanded accordingly to include these computations, hopefully filling the gap in our presentation.
  8. Implemented.
  9. We thank the referee for bringing these results about $\mathbb{R} P^3$ to our attention. We have added a new footnote (number 20) to address this.
  10. We have removed this footnote. [The original purpose was simply to point out that previous considerations of 2d $\mathbb{Z}/2$ global anomalies had considered sets of Majorana-Weyl fermions of a single chirality, whereas our arguments applied more generally to (even numbers of) Majorana-Weyls with arbitrary chiralities. But this should be obvious enough from the main text, and we agree that the footnote is redundant as well as being written rather confusingly.]
  11. We clarified the relationship between $\mathcal{F}$, $f$, and $F$, and added that $\mathcal{F} = \frac{f}{2}\textbf{1}_2 + F$ to the paper. We then noted that the factor of $1/2$ is there to make $f/2\pi$ represent the first Chern class.
  12. The hemisphere $D^2$ should be thought of as one half of $S^2$ with a 2-monopole such that $c_1[S^2] = \int_{S^2}f/(2\pi) = 2$. We clarified this point in the paper as the Referee suggested and also added a few more words to explain the extension of the mapping torus more explicitly.
  13. Already covered in point 4 above.
  14. Implemented.
  15. Amended.

---

## Round 2 · Referee Report · Anonymous (Referee 2) · 2021-2-10

Report

In this paper the authors give a modern reformulation (using bordism) of the old "Elitzur-Nair" method for computing global anomalies by embedding into a bigger groups with local anomalies. This was an important missing result in the recent developments in the subject, and the paper gives a very lucid and clear account of how the old developments should be rephrased in a modern (and better) way.

It is an interesting and well written paper, and I would be happy to recommend publication once the authors clarify some minor points (I am omitting the points that the other referee has already brought up in their review):

  1. In pg. 10, in the last paragraph, it should be "... the vanishing of $Ext^1(\Omega_{d+1}^{H_2}, \mathbb{Z})$ implies $Ext^1(\Omega_{d+1}^{H_2}=0$..." (so $H_1\to H_2$).

  2. At many points in the paper (starting in pg. 23, as far as I can tell) the authors write $H^6_{I\mathbb{Z}}(H)$ with $H$ the symmetry group, instead of $H^6_{I\mathbb{Z}}(MTH)$. Are these two the same object? If not, I would encourage the authors to switch to the second form everywhere (I think that the most natural interpretation of the first form would be the homology of the suspension spectrum of $H$, which is presumably not what the authors intend).

  3. In the beginning of section 6 the authors write "To our knowledge, [ElitzurNair] was the first instance in which a local anomaly was essentially 'pulled back' to derive a global anomaly,...". In fact, Elitzur and Nair credit Witten with the initial idea, in his "Global Aspects of Current Algebra paper". It might be fair to cite this reference too.

  • validity: -
  • significance: -
  • originality: -
  • clarity: -
  • formatting: -
  • grammar: -

Author:  Nakarin Lohitsiri  on 2021-03-02  [id 1280]

(in reply to Report 2 on 2021-02-10)
Category:
answer to question
correction

  1. Amended.
  2. We have fixed the notation as the Referee suggests -- indeed we intended $H_{I\mathbb{Z}}^6(MTH)$ and similar.
  3. We have credited Witten with the earlier suggestion to 'embed' the $SU(2)$ anomaly inside $SU(3)$, both in the introduction (upon first mentioning the paper of Elitzur and Nair) and again at the beginning of Section 6.

---

## Round 2 · Referee Report · Anonymous (Referee 3) · 2021-2-16

Report

This paper is a valuable addition to the growing field of knowledge characterizing anomalies of quantum field theories in terms of mathematical concepts like bordism etc. I wholeheartedly recommend the publication of this paper in SciPost. I have a small complaint however:

In footnote 1, the authors say that the anomaly theory need not be invertible, citing the example of 6d (2,0) theory. Though this might be a sensible statement from the mathematical definition of "anomaly theory" used by the authors, this is not completely sensible from the standard definition used in physics literature. A 't Hooft anomaly is characterized by anomalous phase of the partition function in the presence of non-trivial symmetry background. However, the partition function is well-defined when there is no symmetry background turned on. On the other hand, for the (2,0) theory the partition function with no symmetry background (for the 2-form symmetry) turned on itself is ill-defined. Thus the (2,0) theory has a problem "without the introduction of symmetry into the system" and this is not traditionally referred to as an anomaly. The authors might want to consider clarifying this in the introduction and/or the footnote.
  • validity: -
  • significance: -
  • originality: -
  • clarity: -
  • formatting: -
  • grammar: -

Author:  Nakarin Lohitsiri  on 2021-03-02  [id 1281]

(in reply to Report 3 on 2021-02-16)
Category:
answer to question
correction

We thank the referee for their helpful discussion of the 6d $(2,0)$ anomaly theory. We have rewritten footnote 1 to elaborate on the consequences of an anomaly theory being non-invertible, and why this wouldn't traditionally be thought of as an anomaly from the physics perspective.

---

## Round 3 · Author Response

We thank the referees for their valuable suggestions. We have considered them and revised our manuscript accordingly. Changes specific to them are listed directly as responses to the referees' comments.

---

## Round 3 · List of Changes

- Added a figure (Figure 1) depicting our extension of the $U(2)$ mapping torus to a 6-manifold it bounds.
- Elaborated on the failure of surjectivity in the 2d example (end of section 4).
- Added three unnumbered subsections inside sections 4 and 5.1 for readability.
- Removed the numbering of the subsection 7.2.1.
- Added an explicit specification in the titles of Appendices A.3 and A.4 and in the captions of tables 1 and 2 that the results are valid when m>1.
- See the responses to the referees' comments for other changes.

---

## Editorial Decision

published